# MEASUREMENT SCORE-BASED DIFFUSION MODEL

**Chicago Y. Park**[1]     **Shirin Shoushtari**[2]     **Hongyu An**[2]     **Ulugbek S. Kamilov**[1]
[1]University of Wisconsin–Madison     [2]Washington University in St. Louis
kamilov@wisc.edu

## ABSTRACT

Diffusion models have achieved remarkable success in tasks ranging from image generation to inverse problems. However, training diffusion models typically requires clean ground-truth images, which are unavailable in many applications. We introduce the *Measurement Score-based diffusion Model (MSM)*, a novel framework that learns *partial* measurement scores directly from noisy and subsampled measurements. By aggregating these scores in expectation, MSM synthesizes fully-sampled measurements without requiring access to clean images. To make this practical, we develop a *stochastic sampling* variant of MSM that approximates the expectation efficiently and analyze its asymptotic equivalence to the exact formulation. We further extend MSM to posterior sampling for linear inverse problems, enabling accurate image reconstruction directly from partial scores. Experiments on natural images and multi-coil MRI demonstrate that MSM achieves state-of-the-art performance in unconditional generation and inverse problem solving—all while being trained exclusively on degraded measurements. Code is available at https://github.com/wustl-cig/MSM.

## 1 INTRODUCTION

Score-based diffusion models are powerful generative methods that sample from high-dimensional distributions by learning the score function—the gradient of the log-density—from training data. They achieve state-of-the-art performance in generating natural images (Dhariwal & Nichol, 2021), medical images (Khader et al., 2023), and more (Chung et al., 2024). Beyond generation, diffusion models can be adapted for conditional sampling to solve inverse problems. However, training requires a large set of clean data, which is often costly or difficult to obtain, such as from hardware limits for high-resolution images or long MRI scan times that cause patient discomfort.

To overcome the need for clean training data, recent approaches train diffusion models on subsampled (Daras et al., 2024c), noisy (Xiang et al., 2023; Daras et al., 2024b), or jointly subsampled and noisy observations (Kawar et al., 2024), aiming to approximate the *clean image score* from degraded measurements. However, this is an unnecessarily difficult objective: since measurements typically lie in a structured subspace, recovering the *full image score* from degraded data is inherently challenging.

A more natural strategy is to learn *partial measurement scores* directly within these subspaces. This perspective is closely related to the success of *patch-based methods* in imaging, where training on local patches improves scalability (Alkinani & El-Sakka, 2017; Wang et al., 2023b; Hu et al., 2024a). Our work extends this principle from the image domain to the measurement domain, enabling a new framework for training diffusion models entirely from degraded measurements. Operating in the measurement domain brings a key advantage: each subsampled measurement is uniquely defined by the acquisition operator, whereas a corrupted image is not—infinitely many images can map to the same measurement. Training in the measurement domain removes this ambiguity and ensures that the model learns to denoise well-defined, physically meaningful inputs.

We introduce the *Measurement Score-based diffusion Model (MSM)*, which extends the idea of patch-based learning from the image domain to the measurement domain. Instead of recovering full-image scores, MSM learns denoising score functions restricted to the observable regions of noisy, subsampled measurements—enabling self-supervised training without clean data. By aggregating these partial scores through randomized subsampling, MSM defines an effective framework for both generating full measurements and solving inverse problems. To make MSM practical, we propose

efficient stochastic sampling algorithms for unconditional generation and posterior sampling in linear inverse problems. Extensive experiments on natural images and multi-coil MRI demonstrate that MSM matches or surpasses state-of-the-art performance.

## 2 BACKGROUND

### 2.1 SCORE-BASED DIFFUSION MODELS

Score-based diffusion models (Song & Ermon, 2019; Ho et al., 2020; Song et al., 2021c; Park et al., 2025) learn the score function using neural networks. Tweedie's formula (Efron, 2011) relates the score function to the minimum mean square error (MMSE) denoiser, allowing the score to be estimated using only noisy inputs and their corresponding denoised outputs. Learning score function is performed across varying noise levels, by considering noisy images $x_t = x + \sigma_t n$, where $x$ is a clean image, $n \sim \mathcal{N}(0, I)$, and $\sigma_t$ is the noise level at the timestep $t$.

Given a denoiser $D_\theta$ trained to minimize the mean squared error (MSE), Tweedie's formula approximates the score function as $\nabla \log p_{\sigma_t}(x_t) = (D_\theta(x_t) - x_t)/\sigma_t^2$. This relationship allows denoisers to serve as practical estimators of the score function at varying noise levels, providing gradients that guide reverse-time stochastic sampling (Robbins, 1956; Miyasawa, 1961; Vincent, 2011).

Sampling then proceeds through a sequence of random walks (Song et al., 2021c; Park et al., 2025) as $x_{t-1} = x_t + \tau_t \nabla \log p_{\sigma_t}(x_t) + \sqrt{2\tau_t \mathcal{T}_t} n$, for $t = T, T-1, \ldots, 1$, where $\sigma_t$, $\tau_t$, and $\mathcal{T}_t$ denote the noise level, step-size, and temperature parameters. These parameters can be derived from theoretical frameworks (Ho et al., 2020; Song et al., 2021c) or tuned empirically (Park et al., 2025), and the initial sampling iterate $x_T$ is drawn from a standard Gaussian to be consistent with the training input of the denoiser.

### 2.2 TRAINING DIFFUSION MODELS WITHOUT CLEAN DATA

Training diffusion models to learn the score of clean images typically requires access to high-quality, clean data. However, in many applications, data is often subsampled, noisy, or both.

**Training with noiseless but subsampled measurements.** Ambient diffusion (Daras et al., 2024c; Aali et al., 2025) is a recent method for training diffusion models from subsampled measurements by applying an additional subsampling operation during training. At each step, the model receives a further subsampled and noise-perturbed input and learns to reconstruct the original subsampled measurement. This procedure jointly encourages denoising and inpainting, guiding the model to approximate the conditional expectation of the clean image given a noisy, partially observed input.

**Training with noisy but fully-sampled measurements.** SURE-score (Aali et al., 2023) enables training score-based diffusion models without access to clean data by leveraging Stein's unbiased risk estimator (SURE) (Stein, 1981). SURE-score trains the diffusion model using only noisy measurements, combining two loss functions: a SURE-based loss for denoising the measurements and a denoising loss for the diffusion noise added on top of the denoised estimate. Another approach (Daras et al., 2024b) considers two regimes based on the relationship between measurement noise and diffusion noise. When the measurement noise is smaller than the diffusion noise, the clean image is estimated using Tweedie's formula by predicting the measurement from the diffusion iterate. Conversely, if measurement noise exceeds diffusion noise, the model is trained with a consistency loss (Daras et al., 2024a), which encourages stable denoising outputs across nearby timesteps by enforcing that predictions remain consistent along the model's reverse trajectory.

**Training with noisy and subsampled measurements.** GSURE diffusion (Kawar et al., 2024) trains diffusion models using only noisy, subsampled data by adapting the Generalized SURE loss (Eldar, 2008) to the diffusion setting. It reformulates the training objective as a projected loss computable without clean images. While this objective function is theoretically shown to be equivalent to the supervised diffusion loss under the assumption that the sampling mask and the denoising error are independent, GSURE diffusion has two limitations: it has so far only been demonstrated in the single-coil setting and extending it to multi-coil MRI remains computationally challenging due to the need for singular value decomposition of the full measurement operator; it also requires the minimum diffusion noise level $\sigma_0$ to match the measurement noise level $\rho$. The latter can severely degrade

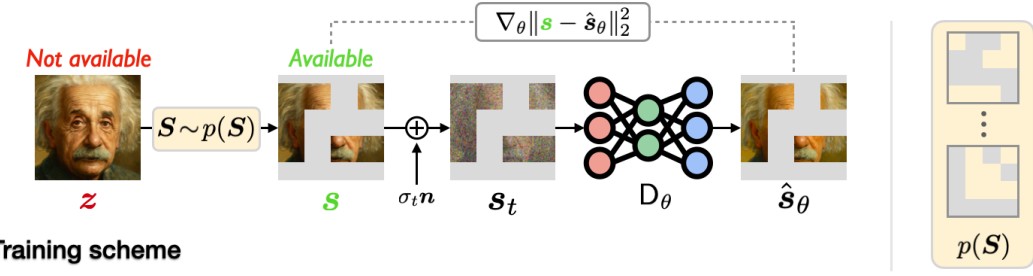

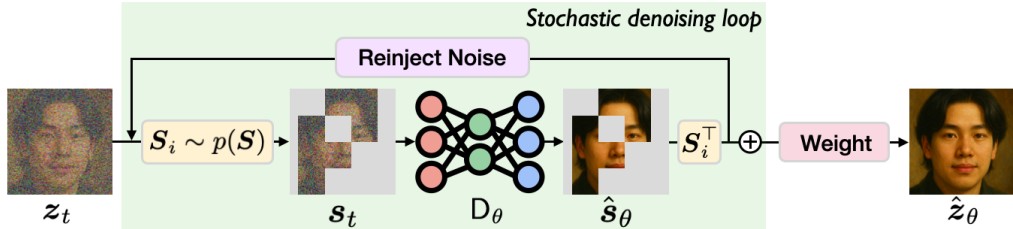

Figure 1: Illustration of the *Measurement Score-based diffusion Model (MSM)* for training and sampling using subsampled data. **Training:** MSM is trained solely on degraded measurements. Diffusion noise is added to these measurements, and the model learns to denoise them. **Sampling:** At each diffusion step, MSM randomly subsamples the current full-measurement iterate, denoises the resulting partial measurement, and aggregates multiple outputs. A weighting vector compensates for overlapping contributions across partial measurements. See Figure 4 for the MRI-specific version.

sampling performance when $\rho$ exceeds typical value of $\sigma_0$ (e.g., $\sigma_0 = 0.01$ in (Ho et al., 2020; Dhariwal & Nichol, 2021)). Other recent methods (Bai et al., 2024; 2025) address the same setting by alternating between reconstructing clean images using diffusion priors pretrained on limited clean data and refining the model to learn from noisy, subsampled measurements.

While these methods differ in how they handle subsampling and noise, they share a common challenge: they attempt to directly approximate the full image score from degraded data. In contrast, our approach leverages the partial measurement-based statistics to generate the full measurements by directly learning measurement scores restricted to observed measurements. A detailed comparison of our approach with related approaches is provided in Appendix E.1.

## 2.3 Imaging Inverse Problems

Inverse problems aim to recover an unknown image $x \in \mathbb{R}^p$ from noisy, undersampled measurements $y \in \mathbb{R}^m$, modeled as $y = Ax + e$, where $A \in \mathbb{R}^{m \times p}$ is a known forward operator and $e \sim \mathcal{N}(0, \eta I)$ is Gaussian noise with variance $\eta$.

When clean training data is unavailable, *self-supervised learning* is often used for training neural networks directly on degraded measurements, without the need for clean ground-truth data (Chen et al., 2021; 2022; Hu et al., 2024b; Yaman et al., 2020; Millard & Chiew, 2023). For example, Self-Supervision via Data Undersampling (SSDU) (Yaman et al., 2020; Millard & Chiew, 2023) is a widely-used approach for training end-to-end restoration networks using distinct subsets of the measurements. In the context of score-based diffusion models, Ambient Diffusion Posterior Sampling (A-DPS) (Aali et al., 2025) replaces the clean-image-trained model in the DPS method (Chung et al., 2023) with an Ambient diffusion model trained on subsampled measurements.

## 3 Measurement Score-Based Diffusion Model

We present the training and sampling procedures for MSM. Our approach generalizes the idea of patch-based learning—widely used to enable computationally efficient supervised training—to self-supervised learning in the measurement domain. A key feature of MSM is that it operates solely on

subsampled measurements during training. This enables learning partial measurement scores without access to clean ground-truth data and naturally extends self-supervised denoising to the challenging setting of noisy, subsampled measurements. We also introduce a conditional sampling algorithm that uses the pretrained MSM to solve linear inverse problems.

## 3.1 LEARNING PARTIAL MEASUREMENT SCORES

We first consider the setting where MSM has access only to partial, noiseless observations of an unknown fully-sampled measurement $\boldsymbol{z} \in \mathbb{R}^n$. Let $\boldsymbol{s} \in \mathbb{R}^m$ denote subsampled measurement as

$$\boldsymbol{s} = \boldsymbol{S}\boldsymbol{z} \in \mathbb{R}^m,$$

where $\boldsymbol{S} \in \{0, 1\}^{m \times n}$, with $m < n$, is a subsampling mask drawn from distribution $p(\boldsymbol{S})$. We assume that in the absence of noise, the fully-sampled measurement $\boldsymbol{z}$ uniquely determines the underlying image $\boldsymbol{x}$. The definition of $\boldsymbol{z}$ depends on the application: for natural images, we set $\boldsymbol{z} = \boldsymbol{x}$, where $\boldsymbol{x} \in \mathbb{R}^p$ is the clean image; for MRI, we set $\boldsymbol{z} = \boldsymbol{F}\boldsymbol{C}\boldsymbol{x}$, where $\boldsymbol{F}$ is the Fourier transform and $\boldsymbol{C}$ is coil-sensitivity operator (Fessler, 2020). More broadly, we suppose that fully-sampled measurements are of form $\boldsymbol{z} = \boldsymbol{T}\boldsymbol{x}$, where $\boldsymbol{T} \in \mathbb{R}^{n \times p}$ ($p \geq n$) is an invertible transformation. It is important to note that MSM is designed to operate solely on partial measurements $\boldsymbol{s} = \boldsymbol{S}\boldsymbol{z}$, with $\boldsymbol{S} \sim p(\boldsymbol{S})$, without access to any fully-sampled $\boldsymbol{z}$ or clean image $\boldsymbol{x}$.

We define a forward diffusion process that adds Gaussian noise to the subsampled measurement $\boldsymbol{s}$

$$\boldsymbol{s}_t = \boldsymbol{s} + \sigma_t \boldsymbol{n}, \quad \boldsymbol{n} \sim \mathcal{N}(\boldsymbol{0}, \boldsymbol{I}),$$

where $t \in \{1, \ldots, T\}$, $\boldsymbol{s}_0 = \boldsymbol{s}$, and $\boldsymbol{s}_T$ approaches a known distribution such as a standard Gaussian. At each step $t$, the diffusion model $\mathsf{D}_\theta$ takes as input the noisy subsampled measurement $\boldsymbol{s}_t$ and is conditioned on the noise level $\sigma_t$, and outputs a denoised estimate of the subsampled measurement $\hat{\boldsymbol{s}}_\theta \in \mathbb{R}^m$ as

$$\hat{\boldsymbol{s}}_\theta(\boldsymbol{s}_t \,;\, \sigma_t) = \mathsf{D}_\theta(\boldsymbol{s}_t \,;\, \sigma_t). \tag{1}$$

The model is trained by minimizing the mean squared error (MSE) loss between the predicted and true subsampled measurements:

$$\mathcal{L}(\theta) = \mathbb{E}_{\boldsymbol{s}_t, t} \left[ \|\boldsymbol{s} - \hat{\boldsymbol{s}}_\theta(\boldsymbol{s}_t \,;\, \sigma_t)\|_2^2 \right].$$

Once trained, the partial measurement score function can be approximated using Tweedie's formula (Efron, 2011), which estimates the gradient of the log-probability of the measurement iterate:

$$\mathsf{S}_\theta(\boldsymbol{s}_t \,;\, \sigma_t, \boldsymbol{S}) = \frac{1}{\sigma_t^2} (\hat{\boldsymbol{s}}_\theta(\boldsymbol{s}_t \,;\, \sigma_t) - \boldsymbol{s}_t), \tag{2}$$

where $\mathsf{S}_\theta \in \mathbb{R}^m$ denotes the learned partial measurement score, explicitly conditioned on the known subsampling mask $\boldsymbol{S}$ that generated $\boldsymbol{s}_t$. Illustrations for training are provided in Figure 1 and Figure 4.

## 3.2 UNCONDITIONAL SAMPLING WITH MSM

Implementing a diffusion model on the fully-sampled measurement requires access to the score function $\nabla \log p_{\sigma_t}(\boldsymbol{z}_t)$, where $\boldsymbol{z}_t$ denotes the noisy version of the fully-sampled measurement $\boldsymbol{z}$. Instead, we train our model to approximate the partial measurement score $\nabla \log p_{\sigma_t}(\boldsymbol{s}_t)$.

The goal of MSM sampling is to generate a fully-sampled measurement $\boldsymbol{z}$ given the partial measurement scores in (2) for $\boldsymbol{S} \sim p(\boldsymbol{S})$. To that end, we define the *MSM score* as the expectation over all possible mask-conditioned partial scores

$$\nabla \log q_{\sigma_t}(\boldsymbol{z}_t) := \boldsymbol{W} \, \mathbb{E}_{\boldsymbol{S} \sim p(\boldsymbol{S})} \left[ \boldsymbol{S}^\top \nabla \log p_{\sigma_t}(\boldsymbol{s}_t \mid \boldsymbol{S}) \Big|_{\boldsymbol{s}_t = \boldsymbol{S}\boldsymbol{z}_t} \right] \tag{3}$$

where each subsampling operator $\boldsymbol{S} \in \mathbb{R}^{m \times n}$ is drawn from distribution $p(\boldsymbol{S})$, and $\boldsymbol{W} \in \mathbb{R}^n$ is a weighting vector that compensates for overlapping contributions across sampling masks. $\boldsymbol{W}$ is defined as the reciprocal of the expected total coverage:

$$\boldsymbol{W} := \left[ \max \left( \mathbb{E}_{\boldsymbol{S} \sim p(\boldsymbol{S})} \left[ \mathrm{diag}(\boldsymbol{S}^\top \boldsymbol{S}) \right], 1 \right) \right]^{-1}, \tag{4}$$

---

**Algorithm 1** Measurement Score-Based Sampling

---

**Require:** $T, p(\boldsymbol{S}), \{\sigma_t\}_{t=1}^T$

1: Initialize $\boldsymbol{z}_T \sim \mathcal{N}(\boldsymbol{0}, \boldsymbol{I})$, $\hat{\boldsymbol{z}}_\theta \leftarrow \boldsymbol{0}$

2: **for** $t = T$ **to** 1 **do**

3:     **for** $i = 1$ **to** $w$ **do**

4:         $\boldsymbol{S}^{(i)} \sim p(\boldsymbol{S})$,   $\boldsymbol{s}_t^{(i)} \leftarrow \boldsymbol{S}^{(i)} \boldsymbol{z}_t$                Partial score-based denoising

5:         $\hat{\boldsymbol{s}}_\theta^{(i)} \leftarrow \boldsymbol{s}_t^{(i)} + \sigma_t^2 \, \mathsf{S}_\theta(\boldsymbol{s}_t^{(i)}; \sigma_t, \boldsymbol{S}^{(i)})$

6:         $\boldsymbol{s}_t^{(i)} \sim p(\boldsymbol{s}_t^{(i)} \mid \hat{\boldsymbol{s}}_\theta^{(i)})$                     Reinject noise & update iterate

7:         $\boldsymbol{z}_t \leftarrow \boldsymbol{S}^{(i)\top} \boldsymbol{s}_t^{(i)} + (\boldsymbol{I} - \boldsymbol{S}^{(i)\top} \boldsymbol{S}^{(i)}) \boldsymbol{z}_t$

8:     **end for**

9:     $\boldsymbol{C} \leftarrow \sum_{i=1}^w \mathrm{diag}(\boldsymbol{S}^{(i)\top} \boldsymbol{S}^{(i)})$             

10:     $\boldsymbol{W} \leftarrow [\max(\boldsymbol{C}, 1)]^{-1}$               Compute weight for aggregation

11:     $\hat{\boldsymbol{z}}_\theta \leftarrow \boldsymbol{W} \sum_{i=1}^w \boldsymbol{S}^{(i)\top} \hat{\boldsymbol{s}}_\theta^{(i)} + \mathbb{1}_{\boldsymbol{C}=0} \cdot \hat{\boldsymbol{z}}_\theta$      Aggregate partial denoised results

12:     $\boldsymbol{z}_{t-1} \sim p(\boldsymbol{z}_{t-1} \mid \boldsymbol{z}_t, \hat{\boldsymbol{z}}_\theta)$

13: **end for**

14: **return** $\boldsymbol{z}_0$

---

where the maximum is applied elementwise to avoid division by zero in regions not covered by any subsampled measurement. Note that the MSM score in (3) can also be interpreted as the score of a product-of-experts (composite-likelihood) model, in which each expert corresponds to a mask-conditioned partial score $\nabla \log p_{\sigma_t}(\boldsymbol{s}_t \mid \boldsymbol{S})$, and the MSM score is obtained by aggregating these partial scores across random masks.

To efficiently approximate the expectation in (3), we propose a stochastic sampling algorithm that uses a randomly selected subset of partial scores. Specifically, we stochastically sample $w$ sampling masks $\boldsymbol{S}^{(i)} \sim p(\boldsymbol{S})$ for $i = 1, \ldots, w$, where $\boldsymbol{S}^{(i)} \in \mathbb{R}^{m_i \times n}$ denotes a subsampling operator $i$. The corresponding subsampled measurement is obtained as $\boldsymbol{s}_t^{(i)} = \boldsymbol{S}^{(i)} \boldsymbol{z}_t$. An unbiased estimator of the MSM score is then defined as

$$\nabla \log \widehat{q}_{\sigma_t}(\boldsymbol{z}_t) := \boldsymbol{W} \left[ \frac{1}{w} \sum_{i=1}^w \boldsymbol{S}^{(i)\top} \nabla \log p_{\sigma_t}(\boldsymbol{s}_t^{(i)} \mid \boldsymbol{S}^{(i)}) \Big|_{\boldsymbol{s}_t^{(i)} = \boldsymbol{S}^{(i)} \boldsymbol{z}_t} \right], \tag{5}$$

where each transpose operator $\boldsymbol{S}^{(i)\top} \in \mathbb{R}^{n \times m_i}$ maps the partial score from the subsampled measurement space back to the fully-sampled measurement space. Note that the reweighting vector $\boldsymbol{W}$ in (4) can be empirically estimated as: $\boldsymbol{W} \leftarrow \left[ \max\left( \sum_{i=1}^w \mathrm{diag}\left( \boldsymbol{S}^{(i)\top} \boldsymbol{S}^{(i)} \right), 1 \right) \right]^{-1}$.

Concretely, MSM performs sampling by iteratively reconstructing a fully-sampled measurement iterate $\boldsymbol{z}_t$ through a stochastic loop of $w$ partial denoising operations. At each diffusion time $t$, the algorithm draws $w$ random subsampling masks $\boldsymbol{S}^{(i)}$, each revealing a different subset of coordinates. For mask $\boldsymbol{S}^{(i)}$, the corresponding partial measurement $\boldsymbol{s}_t^{(i)} = \boldsymbol{S}^{(i)} \boldsymbol{z}_t$ is denoised using the pretrained MSM model to obtain $\hat{\boldsymbol{s}}_\theta^{(i)}$.

Within this stochastic loop, MSM updates $\boldsymbol{z}_t$ after each partial denoising step by (1) drawing a noisy partial estimate $\boldsymbol{s}_t^{(i)} \sim p(\boldsymbol{s}_t^{(i)} \mid \hat{\boldsymbol{s}}_\theta^{(i)})$ and (2) reinserting the noisy partial estimate into the current iterate: $\boldsymbol{z}_t \leftarrow \boldsymbol{S}^{(i)\top} \boldsymbol{s}_t^{(i)} + \left( \boldsymbol{I} - \boldsymbol{S}^{(i)\top} \boldsymbol{S}^{(i)} \right) \boldsymbol{z}_t$. This update ensures that every subsequent partial denoiser operates on an iterate that already incorporates the information extracted in the previous loops, enabling the $w$ stochastic loops to refine complementary regions of $\boldsymbol{z}_t$.

After completing the $w$ stochastic loops, MSM aggregates all partial denoised estimates to form an MMSE estimate of the fully-sampled measurement:

$$\hat{\boldsymbol{z}}_\theta = \boldsymbol{W} \sum_{i=1}^{w} \boldsymbol{S}^{(i)\top} \hat{\boldsymbol{s}}_\theta^{(i)} + \mathbb{1}_{\boldsymbol{C}=0} \cdot \hat{\boldsymbol{z}}_\theta, \tag{6}$$

where $\boldsymbol{C} = \sum_{i=1}^{w} \mathrm{diag}(\boldsymbol{S}^{(i)\top} \boldsymbol{S}^{(i)})$ records which coordinates were covered, and the indicator term preserves previous values at coordinates that receive no coverage by the $w$ stochastic loops.

The estimate $\hat{\boldsymbol{z}}_\theta$ is treated as the clean prediction for the reverse update $\boldsymbol{z}_{t-1} \sim p(\boldsymbol{z}_{t-1} \mid \boldsymbol{z}_t, \hat{\boldsymbol{z}}_\theta)$, following the standard reverse-diffusion step (Ho et al., 2020, Equation 6). Repeating this process from $t = T$ down to 1 produces a fully-sampled measurement $\boldsymbol{z}_0$, which is mapped to the final output image when relevant. The full sampling procedure is outlined in Algorithm 1, with illustrative examples shown in Figure 1 and Figure 4.

In Appendix A, we provide a theoretical guarantee showing that the distribution $\widehat{q}(\boldsymbol{z})$ obtained using the stochastic MSM score converges to the ideal MSM distribution $q(\boldsymbol{z})$ as the number of stochastic iterations $w$ increases. Specifically, under a bounded-variance assumption on the score estimator, the KL divergence satisfies

$$D_{\mathsf{KL}}(q(\boldsymbol{z}) \parallel \widehat{q}(\boldsymbol{z})) \leq \frac{v^2}{w} C,$$

where $v$ is the bounded-variance constant defined in Assumption 1 and $C$ is a finite constant independent of $w$.

## 3.3 POSTERIOR SAMPLING WITH MSM

We extend MSM to sample from the posterior distribution for solving linear inverse problems described in Section 2.3. We consider measurement operators $\boldsymbol{A}$ of the form $\boldsymbol{A} = \boldsymbol{HT}$, where $\boldsymbol{H} \in \mathbb{R}^{m \times n}$ is the linear measurement operator—such as downsampling, blurring, box inpainting, or random projection—and $\boldsymbol{T} \in \mathbb{R}^{n \times p}$ is an invertible transformation introduced in Section 3.1. This allows us to express the measurement model as $\boldsymbol{y} = \boldsymbol{Hz} + \boldsymbol{e}$, where $\boldsymbol{z} \in \mathbb{R}^n$ is the unknown fully-sampled measurement and $\boldsymbol{e} \sim \mathcal{N}(\boldsymbol{0}, \eta\boldsymbol{I})$ is Gaussian noise with variance $\eta$.

To estimate the posterior score, we combine the stochastic score estimate from (6) with its corresponding fully-sampled prediction $\hat{\boldsymbol{z}}_\theta$. The posterior score is approximated as:

$$\nabla \log p_{\sigma_t}(\boldsymbol{z}_t \mid \boldsymbol{y}) = \nabla \log p_{\sigma_t}(\boldsymbol{z}_t) + \nabla \log p_{\sigma_t}(\boldsymbol{y} \mid \boldsymbol{z}_t)$$
$$\approx \nabla \log \widehat{q}_{\sigma_t}(\boldsymbol{z}_t) + \nabla \log p_{\sigma_t}(\boldsymbol{y} \mid \hat{\boldsymbol{z}}_\theta),$$

where the log-likelihood gradient is given by

$$\nabla \log p_{\sigma_t}(\boldsymbol{y} \mid \hat{\boldsymbol{z}}_\theta) = \gamma_t \nabla \|\boldsymbol{y} - \boldsymbol{H}\hat{\boldsymbol{z}}_\theta\|_2^2, \tag{7}$$

with $\gamma_t$ denoting the step-size parameter. Note that $\boldsymbol{H}$ may differ from the randomized subsampling operators $\boldsymbol{S} \sim p(\boldsymbol{S})$ used during MSM training.

We incorporate the likelihood term by updating the denoised estimate via

$$\hat{\boldsymbol{z}}_\theta \leftarrow \hat{\boldsymbol{z}}_\theta - \gamma_t \nabla_{\hat{\boldsymbol{z}}_\theta} \|\boldsymbol{y} - \boldsymbol{H}\hat{\boldsymbol{z}}_\theta\|_2^2.$$

This update is inserted between lines 11 and 12 in Algorithm 1 to convert unconditional MSM sampling into posterior sampling.

A related posterior sampling strategy, using diffusion models trained on clean data, was proposed in (Wang et al., 2023a). Further simplification of our method for compressed-sensing MRI is presented in Appendix D.3.

## 3.4 LEARNING PARTIAL MEASUREMENT SCORE FROM NOISY AND SUBSAMPLED MEASUREMENTS

We now show how our MSM framework can be extended to train directly on noisy and subsampled measurements by integrating with self-supervised denoising methods to address the noise on the subsampled measurements.

We formulate our observed measurement as

$$\boldsymbol{s} = \boldsymbol{S}\boldsymbol{z} + \boldsymbol{\nu}, \quad \boldsymbol{\nu} \sim \mathcal{N}(\boldsymbol{0}, \rho\boldsymbol{I}),$$

where $\rho$ is the measurement noise level, and the remaining notations follow Section 3.1. We first define the sequence of diffusion noise level $\{\sigma_t\}_{t=1}^T$. At each time step $t$, we compare the diffusion noise level $\sigma_t$ with the measurement noise level $\rho$, and apply one of the following strategies accordingly:

**Case 1:** $\sigma_t > \rho$**.** We add residual noise to match the diffusion level:

$$\boldsymbol{s}_t \leftarrow \boldsymbol{s} + \sqrt{\sigma_t^2 - \rho^2}\boldsymbol{n}.$$

The training objective is:

$$\mathcal{L}(\theta) = \mathbb{E}_{\boldsymbol{s}_t, t}\left[\|\boldsymbol{s} - \mathbb{E}[\boldsymbol{s} \mid \boldsymbol{s}_t]\|_2^2\right] + \mathcal{L}_{\text{SURE}}(\theta\,;\boldsymbol{s}, \rho)$$

$$= \mathbb{E}_{\boldsymbol{s}_t, t}\left[\left\|\boldsymbol{s} - \left(\frac{\sigma_t^2 - \rho^2}{\sigma_t^2}(\hat{\boldsymbol{s}}_\theta(\boldsymbol{s}_t; \sigma_t) - \boldsymbol{s}_t) + \boldsymbol{s}_t\right)\right\|_2^2\right] + \mathcal{L}_{\text{SURE}}(\theta\,;\boldsymbol{s}, \rho),$$

where the first term is inspired by (Daras et al., 2024b), which shows that a noisier image can be denoised using a less noisy reference; we apply this idea to subsampled measurements. The second term is the SURE loss, following (Chen et al., 2022, Equation 9), which enables the model to learn to denoise measurement noise and plays a key role in the next case.

**Case 2:** $\sigma_t \le \rho$**.** We first denoise $\boldsymbol{s}$ using the MSM with the noise conditioned of $\rho$, then add diffusion noise as

$$\boldsymbol{s}_t \leftarrow \hat{\boldsymbol{s}}_\theta(\boldsymbol{s}\,;\rho) + \sigma_t\boldsymbol{n}.$$

Training minimizes the discrepancy within the non-subsampled region:

$$\mathcal{L}(\theta) = \mathbb{E}_{\boldsymbol{s}_t, t}[\hat{\boldsymbol{s}}_\theta(\boldsymbol{s}\,;\rho) - \hat{\boldsymbol{s}}_\theta(\boldsymbol{s}_t\,;\sigma_t)\|_2^2] + \mathcal{L}_{\text{SURE}}(\theta\,;\boldsymbol{s}, \rho).$$

Here, $\hat{\boldsymbol{s}}_\theta(\boldsymbol{s}; \rho)$ serves as a pseudo-clean reference. Its quality is crucial but improves naturally during training, since the same prediction is refined in **Case 1**. In practice, most $\sigma_t$ are larger than $\rho$, making **Case 1** more frequently sampled. As a result, the pseudo-clean reference used in **Case 2** is continuously improved, ensuring stable training across both cases.

## 4    NUMERICAL EVALUATIONS

We evaluated MSM on unconditional generation and conditional sampling for natural images and multi-coil MRI. All models used the same diffusion architecture (Dhariwal & Nichol, 2021), trained from scratch on a single NVIDIA A100 GPU for 1M iterations (see Appendix D.4 for architectural details). Experiments used 69k FFHQ faces ($128 \times 128$ RGB) and 2k fastMRI T2-weighted slices ($256 \times 256$, complex-valued) (Zbontar et al., 2018; Knoll et al., 2020). We performed inverse problem evaluations on 100 test images per domain.

### 4.1    RGB FACE IMAGE EXPERIMENT

**Training data.** We considered two training settings: (1) subsampling only and (2) subsampling with added Gaussian noise $\rho = 0.1$. In both cases, the masked-pixel ratio $p$ is set to 40% using $32 \times 32$ patches, applied identically across RGB channels (see Figure 2 for an example). Full training details of our method and baselines are provided in Appendix D.4 and Appendix D.5.

**Unconditional sampling.** MSM used a stochastic loop parameter $w = 1$ with 200 sampling steps. We compared MSM to three baselines: an oracle diffusion model trained on clean images, Ambient diffusion (Daras et al., 2024c) trained on noiseless masked inputs, and GSURE diffusion (Kawar et al., 2024) trained on noisy masked inputs. All baselines used 200-step accelerated sampling (Song et al., 2021a). As shown in Table 1, MSM achieves better FID scores than all baselines trained without clean data, evaluated over 10,000 generated samples. Figure 2 further shows that MSM generates clean images despite being trained without clean data. Additional results showing how $w$ influences sampling quality and time efficiency are provided in Appendix E.6.

Table 1: FID scores for unconditional image samples under different training scenarios on human face images. **Best values** are highlighted for each training scenario, with comparisons shown when corresponding baseline methods are available. Note how MSM consistently achieves substantially lower FID scores than alternative methods across the evaluated settings.

| Training data | Methods | FID↓ |
|---|---|---|
| No degradation | Oracle diffusion | 10.21 |
| $p = 0.4, \rho = 0$ | MSM | **29.14** |
| | Ambient diffusion | 55.90 |
| $p = 0.4, \rho = 0.1$ | MSM | **37.14** |
| | GSURE diffusion | 89.71 |

Table 2: FID scores for unconditional image samples under different training scenarios on multi-coil brain MR images. **Best values** are highlighted for each training scenario, with comparisons shown when corresponding baseline methods are available. Note how MSM achieves a lower FID compared to the alternative methods.

| Training data | Methods | FID↓ |
|---|---|---|
| No degradation | Oracle diffusion | 28.41 |
| $R = 4, \rho = 0$ | MSM | **64.37** |
| | Ambient diffusion | 70.07 |
| $R = 4, \rho = 0.1$ | MSM | 82.17 |

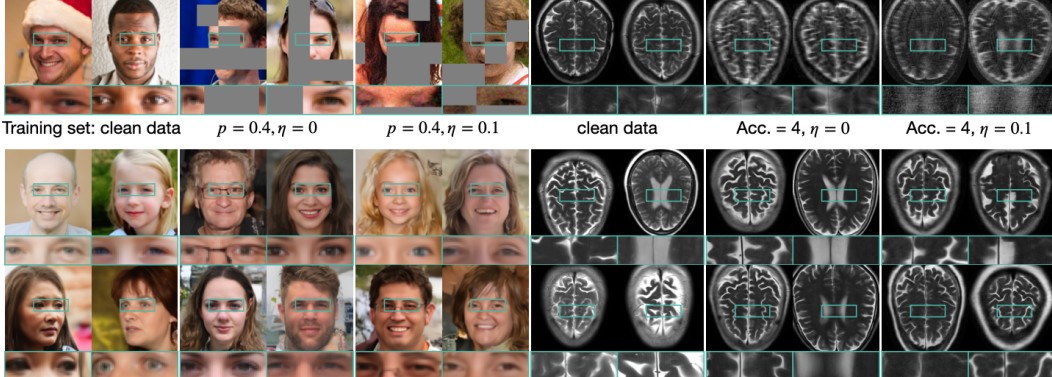

Training set: clean data   $p = 0.4, \eta = 0$   $p = 0.4, \eta = 0.1$   clean data   Acc. = 4, $\eta = 0$   Acc. = 4, $\eta = 0.1$

Generated samples

Figure 2: Generated samples from MSM trained under three degradation settings (first row: training data; second row: samples generated by models trained on the corresponding data). Note how despite never seeing ground-truth data, MSM can generate high-quality images.

**Conditional sampling.** We compared MSM with A-DPS (Aali et al., 2025), which is the most closely related inverse-problem solver that also uses a pretrained diffusion model trained on incomplete data. We evaluated two inverse problems: box inpainting with a $64 \times 64$ missing region and $4\times$ bicubic super-resolution. Both methods used models trained on the same noiseless 40% masked data without retraining. A-DPS took 1000 steps, while MSM took 200 steps with $w = 3$, and step sizes in (7) set to $\gamma_t = 1.75$ for both inpainting and super-resolution. As shown in Table 3 and Figure 3, MSM outperforms A-DPS in PSNR, SSIM, and LPIPS. We also observe that A-DPS performs worse than the input image in PSNR and SSIM, likely due to limitations of the Ambient diffusion prior in generating fine details when trained on non-sparse subsampling patterns—such as box masks—rather than more localized patterns (e.g., dust masks) mainly used in the original training setup of Ambient diffusion (Daras et al., 2024c). Detailed hyperparameter setups for A-DPS are provided in Appendix D.6. Additional experiments showing that MSM achieves comparable performance to clean-data-trained diffusion-based inverse problem solvers are provided in Appendix E.5. Results using an MSM prior trained on noisy and subsampled data are provided in Appendix E.3.

## 4.2 MULTI-COIL MRI EXPERIMENT

**Training data.** We considered two training settings: (1) subsampling only and (2) subsampling with added Gaussian noise $\rho = 0.1$. We applied a 1-D Cartesian subsampling operation in k-space using random masks with an acceleration rate of $R = 4$, while fully sampling all vertical lines and the central 20 lines for autocalibration. Training details for our method and baselines are provided in Appendix D.4 and D.5.

Table 3: Quantitative results on two natural image inverse problems comparing methods using diffusion priors trained without clean images. **Best values** are highlighted per metric. MSM achieves the best performance across both distortion-based and perception-oriented metrics.

Table 4: Quantitative results on multi-coil compressed sensing MRI comparing diffusion-based and self-supervised methods, all trained without clean data. **Best values** are highlighted per metric. MSM outperforms the baselines in both PSNR and LPIPS, including the restoration-specific baseline SSDU.

| Setup | | Input | A-DPS | MSM |
|---|---|---|---|---|
| | PSNR↑ | 18.26 | 20.14 | **24.71** |
| **Inpainting** | SSIM↑ | 0.749 | 0.621 | **0.867** |
| | LPIPS↓ | 0.304 | 0.305 | **0.076** |
| | PSNR↑ | 23.21 | 22.61 | **28.11** |
| **SR** (×4) | SSIM↑ | 0.728 | 0.702 | **0.868** |
| | LPIPS↓ | 0.459 | 0.277 | **0.117** |

| Setup | | Input | A-DPS | SSDU | MSM |
|---|---|---|---|---|---|
| | PSNR↑ | 22.75 | 27.28 | 29.65 | **30.71** |
| **CS-MRI** (×4) | SSIM↑ | 0.648 | 0.804 | **0.847** | 0.839 |
| | LPIPS↓ | 0.306 | 0.173 | 0.160 | **0.145** |
| | PSNR↑ | 21.94 | 26.29 | 28.02 | **28.86** |
| **CS-MRI** (×6) | SSIM↑ | 0.617 | 0.763 | **0.820** | 0.805 |
| | LPIPS↓ | 0.342 | 0.201 | 0.186 | **0.168** |

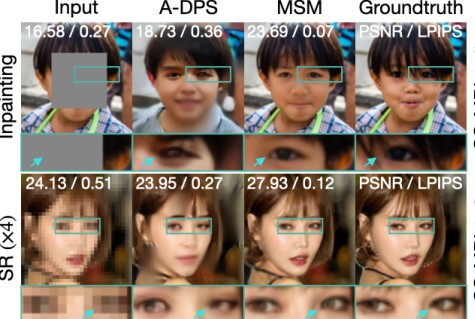 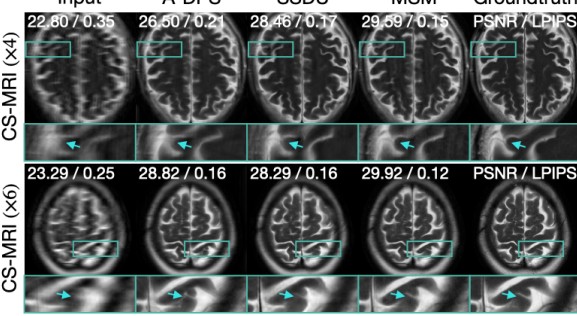

Figure 3: Visual comparison of methods trained on subsampled data for inverse problems. Note how MSM leads to the best results in both applications.

**Unconditional sampling.** MSM used a stochastic loop parameter of $w = 1$ with 200 sampling steps (see Appendix E.6 for how $w$ affects sampling quality). We compared it against an oracle diffusion model trained on clean images and Ambient diffusion (Daras et al., 2024c) trained on noiseless subsampled inputs; GSURE diffusion (Kawar et al., 2024) is omitted since its extension from single-coil MRI to multi-coil MRI remains computationally infeasible and no practical approach has been proposed. All baselines used accelerated sampling with 200 steps (Song et al., 2021a). As shown in Table 2, MSM achieves better FID scores than Ambient diffusion, based on 10,000 generated samples. Figure 2 further demonstrates that MSM generates realistic images, even when trained on noisy and subsampled measurements. FID computation details are provided in Appendix D.7, and generation results under extreme subsampling are shown in Appendix E.2.

**Conditional sampling.** We evaluated accelerated MRI reconstruction using random masks with acceleration rates $R = 4$ and $R = 6$, and measurement noise $\eta = 0.01$. We used stochastic posterior sampling algorithm introduced in Appendix D.3 with MSM pretrained on noiseless $R = 4$ measurements without retraining, with step size $\gamma_t = 2$ in (19). We compared against two baselines trained on the same subsampled data: a diffusion-based method (A-DPS (Aali et al., 2025)) and a self-supervised end-to-end method (Robust SSDU (Millard & Chiew, 2024)). A-DPS used 1000 steps, SSDU performed a single forward pass, and MSM used 200 steps with $w = 3$. As shown in Table 4 and Figure 3, MSM outperforms both baselines in PSNR and LPIPS. Detailed hyperparameter setups for A-DPS and Robust SSDU are provided in Appendix D.6. Comparisons with clean-data-trained diffusion-based inverse problem solvers are in Appendix E.5, and additional results using an MSM prior trained on noisy and subsampled MRI data are in Appendix E.3.

## 5 CONCLUSION

We introduced the *Measurement Score-based diffusion Model (MSM)* framework for generating the full measurements using score functions learned solely from noisy, subsampled measurements. Our

key idea is the MSM score, defined as an expectation over partial scores induced by randomized subsampling. We develop a stochastic sampling algorithm for both prior and posterior inference that efficiently approximates this expectation, enabling clean image generation and inverse problem solving. We demonstrate that MSM achieves state-of-the-art performance among diffusion-based methods trained without clean data, for both unconditional image generation and conditional sampling in linear inverse problems. The framework applies broadly to settings where only subsampled measurements are available but collectively cover the full data space, making it valuable for generative modeling in limited-data regimes and high-dimensional sampling from low-dimensional observations.

## 6 ACKNOWLEDGMENTS

This work was supported in part by the National Science Foundation under Grants No. 2504613 and No. 2043134 (CAREER), and by the National Institutes of Health (NIH) under Grants R01EB032713 and RF1NS116565.

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

## A   THEORETICAL ANALYSIS

We have introduced the MSM framework, which can generate fully-sampled measurements using *partial measurement scores*. To make the expectation of the MSM score (3) efficient, our algorithm approximates it using a minibatch of $w$ sampled operators in (5), where $\boldsymbol{S}^{(1)}, \ldots, \boldsymbol{S}^{(w)}$ are sampled independently and identically from the distribution $p(\boldsymbol{S})$. This implies that for a fixed $\boldsymbol{W}$, we have an unbiased estimator of the MSM score: $\mathbb{E}\left[\nabla \log \widehat{q}_{\sigma_t}(\boldsymbol{z}_t)\right] = \nabla \log q_{\sigma_t}(\boldsymbol{z}_t)$, where the expectation is over the randomness in the sampled minibatch.

**Assumption 1.** *There exists $v > 0$ such that for all $\boldsymbol{z} \in \mathbb{R}^n$,*

$$\mathbb{E}\left[\|\nabla \log q_{\sigma_t}(\boldsymbol{z}) - \nabla \log \widehat{q}_{\sigma_t}(\boldsymbol{z})\|_2^2\right] \leq \frac{v^2}{w},$$

*where the expectation is taken over $\boldsymbol{S}^{(i)} \sim p(\boldsymbol{S})$.*

This assumption implies that the gradient estimate of MSM score has a bounded variance, an assumption commonly adopted in stochastic and online algorithms (Welling & Teh, 2011; Ghadimi & Lan, 2016; Liu et al., 2022).

**Theorem 1.** *Let $q(\boldsymbol{z})$ and $\widehat{q}(\boldsymbol{z})$ denote the distributions of samples generated by using the MSM score $\nabla \log q_{\sigma_t}(\boldsymbol{z}_t)$ and its stochastic approximation $\nabla \log \widehat{q}_{\sigma_t}(\boldsymbol{z}_t)$, respectively. Under Assumption 1, the KL divergence between the two distributions is bounded as*

$$D_{\mathsf{KL}}(q(\boldsymbol{z}) \parallel \widehat{q}(\boldsymbol{z})) \leq \frac{v^2}{w}C,$$

*where $C$ is a finite constant independent of $w$.*

*Proof.* Our proof invokes Girsanov's theorem, which characterizes how the distribution of a Brownian-driven stochastic process transforms when we transition from one probability measure to another (Chen et al., 2024; Baker et al., 2024; Huang et al., 2021; Song et al., 2021b).

Consider the two stochastic processes $\{\boldsymbol{z}(t)\}_{t \in [0,1]}$ and $\{\widehat{\boldsymbol{z}}(t)\}_{t \in [0,1]}$, corresponding to the Euler–Maruyama discretizations of the following reverse-time SDEs

$$\begin{aligned}
\mathrm{d}\boldsymbol{z} &= \nabla \log q_{\sigma_t}(\boldsymbol{z})\mathrm{d}t + \sqrt{2\mathcal{T}_t}\mathrm{d}\overline{\boldsymbol{w}}_t & \boldsymbol{z}(T) &= \boldsymbol{z}_T \sim \mathcal{N}(\boldsymbol{0}, \boldsymbol{I}), \\
\mathrm{d}\widehat{\boldsymbol{z}} &= \nabla \log \widehat{q}_{\sigma_t}(\widehat{\boldsymbol{z}})\mathrm{d}t + \sqrt{2\mathcal{T}_t}\mathrm{d}\overline{\boldsymbol{w}}_t & \widehat{\boldsymbol{z}}(T) &= \widehat{\boldsymbol{z}}_T \sim \mathcal{N}(\boldsymbol{0}, \boldsymbol{I}),
\end{aligned}$$

Let $\mathbb{Q}[\boldsymbol{z}_{0:T}]$ and $\widehat{\mathbb{Q}}[\widehat{\boldsymbol{z}}_{0:T}]$ denote the path measures induced by the respective processes. The process $\{\boldsymbol{z}(t)\}_{t \in [0,1]}$ corresponds to the reverse diffusion trajectory driven by the true measurement score $\nabla \log q_{\sigma_t}(\boldsymbol{z})$, while $\{\widehat{\boldsymbol{z}}(t)\}_{t \in [0,1]}$ is generated by the reverse process using an approximate (stochastic) measurement score $\nabla \log \widehat{q}_{\sigma_t}(\boldsymbol{z})$.

By using the chain rule of KL divergence from Lemma 3, we have

$$D_{\mathsf{KL}}(\mathbb{Q} \parallel \widehat{\mathbb{Q}}) = \mathbb{E}_{\boldsymbol{z}_T \sim \mathcal{N}(\boldsymbol{0}, \boldsymbol{I})}\left[D_{\mathsf{KL}}(\mathbb{Q}(.|\boldsymbol{z} = \boldsymbol{z}_T) \parallel \widehat{\mathbb{Q}}(.|\widehat{\boldsymbol{z}} = \widehat{\boldsymbol{z}}_T))\right]. \tag{8}$$

Using the definition of KL divergence and the fact that $M_T = \mathrm{d}\widehat{\mathbb{Q}}/\mathrm{d}\mathbb{Q}$ from Lemma 1, we have

$$D_{\mathsf{KL}}(\mathbb{Q}(.|\boldsymbol{z} = \boldsymbol{z}_T) \parallel \widehat{\mathbb{Q}}(.|\widehat{\boldsymbol{z}} = \widehat{\boldsymbol{z}}_T)) = -\mathbb{E}_{\mathbb{Q}}\left[\log \frac{\mathrm{d}\widehat{\mathbb{Q}}}{\mathrm{d}\mathbb{Q}}\right] = -\mathbb{E}_{\mathbb{Q}}\left[\log M_T\right]$$

$$= \mathbb{E}_{\mathbb{Q}}\left[\int_0^T \frac{[\nabla \log q_{\sigma_t}(\boldsymbol{z}_t) - \nabla \log \widehat{q}_{\sigma_t}(\boldsymbol{z}_t)]}{\sqrt{2\mathcal{T}_t}} \mathrm{d}\overline{\boldsymbol{w}}_t \quad + \frac{1}{2}\int_0^T \frac{\|\nabla \log q_{\sigma_t}(\boldsymbol{z}_t) - \nabla \log \widehat{q}_{\sigma_t}(\boldsymbol{z}_t)\|_2^2}{2\mathcal{T}_t} \mathrm{d}t\right]$$

$$= \mathbb{E}_{\mathbb{Q}}\left[\int_0^T \frac{[\nabla \log q_{\sigma_t}(\boldsymbol{z}_t) - \nabla \log \widehat{q}_{\sigma_t}(\boldsymbol{z}_t)]}{\sqrt{2\mathcal{T}_t}} \mathrm{d}\overline{\boldsymbol{w}}_t\right] + \mathbb{E}_{\mathbb{Q}}\left[\frac{1}{2}\int_0^T \frac{\|\nabla \log q_{\sigma_t}(\boldsymbol{z}_t) - \nabla \log \widehat{q}_{\sigma_t}(\boldsymbol{z}_t)\|_2^2}{2\mathcal{T}_t} \mathrm{d}t\right]$$

$$= \mathbb{E}_{\mathbb{Q}}\left[\int_0^T \frac{1}{\sqrt{2\mathcal{T}_t}} \mathbb{E}\left[\nabla \log q_{\sigma_t}(\boldsymbol{z}_t) - \frac{1}{w}\sum_{i=1}^w \nabla \log q_{\sigma_t}(\boldsymbol{s}_t^{(i)})\Big|_{\boldsymbol{s}_t^{(i)}=\boldsymbol{S}^{(i)}\boldsymbol{z}_t}\right] \mathrm{d}\overline{\boldsymbol{w}}_t\right]$$

$$+ \mathbb{E}_{\mathbb{Q}}\left[\int_0^T \frac{1}{4\mathcal{T}_t} \mathbb{E}\left[\left\|\nabla \log q_{\sigma_t}(\boldsymbol{z}_t) - \frac{1}{w}\sum_{i=1}^w \nabla \log q_{\sigma_t}(\boldsymbol{s}_t^{(i)})\Big|_{\boldsymbol{s}_t^{(i)}=\boldsymbol{S}^{(i)}\boldsymbol{z}_t}\right\|_2^2\right] \mathrm{d}t\right]$$

$$= \mathbb{E}_{\mathbb{Q}}\left[\int_0^T \frac{1}{4\mathcal{T}_t} \mathbb{E}\left[\left\|\nabla \log q_{\sigma_t}(\boldsymbol{z}_t) - \frac{1}{w}\sum_{i=1}^w \nabla \log q_{\sigma_t}(\boldsymbol{s}_t^{(i)})\Big|_{\boldsymbol{s}_t^{(i)}=\boldsymbol{S}^{(i)}\boldsymbol{z}_t}\right\|_2^2\right] \mathrm{d}t\right] \le \frac{v^2}{w}\int_0^T \frac{1}{4\mathcal{T}_t} \mathrm{d}t \le \frac{v^2}{w}C,$$

where $C \coloneqq \int_0^T 1/(4\mathcal{T}_t)\mathrm{d}t$ is a finite constant. In the first line, we use the definition of KL divergence between $\mathbb{Q}$ and $\widehat{\mathbb{Q}}$ and the result from Lemma 1. In the third line, we use the law of iterated expectations over $w$ sampling masks $\boldsymbol{S}^{(i)} \sim p(\boldsymbol{S})$ for $i = 1, \ldots, w$. Note that since $\mathbb{E}[\nabla \log \widehat{q}_{\sigma_t}(\boldsymbol{z}_t)]$ is an unbiased estimator of MSM score $\nabla \log q_{\sigma_t}(\boldsymbol{z}_t)$, we have $\mathbb{E}[\nabla \log \widehat{q}_{\sigma_t}(\boldsymbol{z}_t)] = \nabla \log q_{\sigma_t}(\boldsymbol{z}_t)$, which yields the expectation in the fourth line to be $0$. In the last line, we use the bounded variance in Assumption 1.

Following this result with (8) and Lemma 2, we have

$$D_{\mathsf{KL}}(q_0 \parallel \widehat{q}_0) \le D_{\mathsf{KL}}(\mathbb{Q} \parallel \widehat{\mathbb{Q}}) \le \frac{v^2}{w}C. \tag{9}$$

$\square$

**Lemma 1.** *(**The Girsanov Theorem III.**) Let $\{\boldsymbol{z}(t)\}_{t=T}^0$ and $\{\widehat{\boldsymbol{z}}(t)\}_{t=T}^0$ be two Itô process of the forms*

$$\mathrm{d}\boldsymbol{z} = \nabla \log q_{\sigma_t}(\boldsymbol{z})\mathrm{d}t + \sqrt{2\mathcal{T}_t}\mathrm{d}\overline{\boldsymbol{w}}_t \quad \boldsymbol{z}(T) = \boldsymbol{z}_T \sim \mathcal{N}(\boldsymbol{0}, \boldsymbol{I})$$
$$\mathrm{d}\widehat{\boldsymbol{z}} = \nabla \log \widehat{q}_{\sigma_t}(\widehat{\boldsymbol{z}})\mathrm{d}t + \sqrt{2\mathcal{T}_t}\mathrm{d}\overline{\boldsymbol{w}}_t \quad \widehat{\boldsymbol{z}}(T) = \widehat{\boldsymbol{z}}_T \sim \mathcal{N}(\boldsymbol{0}, \boldsymbol{I}),$$

*where $0 \le T \le \infty$ is a given constant, and $\overline{\boldsymbol{w}} \in \mathbb{R}^n$ is a $n-$dimensional Brownian motion. Suppose that there exist a process $\alpha(\boldsymbol{z}, t)$ such that*

$$\alpha(\boldsymbol{z}, t) = \frac{[\nabla \log q_{\sigma_t}(\boldsymbol{z}_t) - \nabla \log \widehat{q}_{\sigma_t}(\boldsymbol{z}_t)]}{\sqrt{2\mathcal{T}_t}}, \tag{10}$$

*which satisfies Novikov's condition*

$$\mathbb{E}\left[exp\left(\frac{1}{2}\int_0^T \alpha^2(\boldsymbol{z}, t)\mathrm{d}t\right)\right] = \mathbb{E}\left[exp\left(\frac{1}{2}\int_0^T \frac{\|\nabla \log q_{\sigma_t}(\boldsymbol{z}_t) - \nabla \log \widehat{q}_{\sigma_t}(\boldsymbol{z}_t)\|_2^2}{2\mathcal{T}_t} \mathrm{d}t\right)\right] < \infty,$$

*where $\mathbb{E} = \mathbb{E}_{\mathbb{Q}}$ is the expectation with respect to $\mathbb{Q}$, probability measure induced by the process $\{\boldsymbol{z}_t\}_{t=T}^0$. Then, we can define $M_T$ and probability measure $\widehat{\mathbb{Q}}$, induced by process $\{\widehat{\boldsymbol{z}}_t\}_{t=T}^0$ as $M_T \coloneqq$*

$$exp\left(-\int_0^T \frac{[\nabla \log q_{\sigma_t}(\boldsymbol{z}_t) - \nabla \log \widehat{q}_{\sigma_t}(\boldsymbol{z}_t)]}{\sqrt{2\mathcal{T}_t}} \mathrm{d}\overline{\boldsymbol{w}}_t - \frac{1}{2}\int_0^T \frac{\|\nabla \log q_{\sigma_t}(\boldsymbol{z}_t) - \nabla \log \widehat{q}_{\sigma_t}(\boldsymbol{z}_t)\|_2^2}{2\mathcal{T}_t} \mathrm{d}t\right),$$

*where*

$$t \le T \quad and \quad \mathrm{d}\widehat{\mathbb{Q}} \coloneqq M_T\mathrm{d}\mathbb{Q}. \tag{11}$$

Proof of the Girsanov Theorems can be found in (Oksendal, 2013, Theorems 8.6.3, 8.6.4, and 8.6.5).

**Remark.** Note that it can be shown that Novikov's condition is satisfied for function $\alpha(\boldsymbol{z}, t)$ in (10) as

$$
\mathbb{E}\left[exp\left(\frac{1}{2}\int_0^T \frac{\|\nabla \log q_{\sigma_t}(\boldsymbol{z}_t) - \nabla \log \widehat{q}_{\sigma_t}(\boldsymbol{z}_t)\|_2^2}{2\mathcal{T}_t}\mathrm{d}t\right)\right]
$$

$$
= \mathbb{E}\left[exp\left(\frac{1}{2}\int_0^T \frac{\mathbb{E}\left[\|\nabla \log q_{\sigma_t}(\boldsymbol{z}_t) - \nabla \log \widehat{q}_{\sigma_t}(\boldsymbol{z}_t)\|_2^2\right]}{2\mathcal{T}_t}\mathrm{d}t\right)\right]
$$

$$
\leq \frac{v^2}{w} \cdot exp\left(\frac{1}{2}\int_0^T \frac{1}{2\mathcal{T}_t}\mathrm{d}t\right) < \infty,
$$

where in the second line, we use the total law of expectation (i.e., $\mathbb{E}[\boldsymbol{a}] = \mathbb{E}[\mathbb{E}[\boldsymbol{a}|\boldsymbol{b}]]$) we use the law of iterated expectations over $w$ sampling masks $\boldsymbol{S}^{(i)} \sim p(\boldsymbol{S})$ for $i = 1, \ldots, w$. Here, we use Assumption 1and the fact the $\int_0^T (1/(2\mathcal{T}_t))\,\mathrm{d}t$ is a finite constant.

**Lemma 2.** *Let $\mathbb{Q}$ and $\widehat{\mathbb{Q}}$ be the path measure of two stochastic processes $\{\boldsymbol{z}(t)\}_{t=0}^T$ and $\{\widehat{\boldsymbol{z}}(t)\}_{t=0}^T$. We denote $q_0$ and $\widehat{q}_0$ as the marginal distribution of $\boldsymbol{z}(0)$ and $\widehat{\boldsymbol{z}}(0)$. Then, we have*

$$
D_{\mathsf{KL}}(q_0 \parallel \widehat{q}_0) \leq D_{\mathsf{KL}}(\mathbb{Q} \parallel \widehat{\mathbb{Q}}).
$$

*Proof.* From the chain rule of KL divergence, we have

$$
D_{\mathsf{KL}}(\mathbb{Q} \parallel \widehat{\mathbb{Q}}) = D_{\mathsf{KL}}(\mathbb{Q}_{\boldsymbol{z}(0)=\boldsymbol{z}_0} \parallel \widehat{\mathbb{Q}}_{\widehat{\boldsymbol{z}}(0)=\boldsymbol{z}_0})
$$

$$
+ \int_{\boldsymbol{z}} D_{\mathsf{KL}}(\mathbb{Q}(. \mid \boldsymbol{z}(0) = \boldsymbol{z}_0) \parallel \widehat{\mathbb{Q}}(. \mid \widehat{\boldsymbol{z}}(0) = \boldsymbol{z}_0))\mathbb{Q}_{\boldsymbol{z}(0)=\boldsymbol{z}_0}(\mathrm{d}\boldsymbol{z})
$$

$$
= D_{\mathsf{KL}}(q_0 \parallel \widehat{q}_0) + \int_{\boldsymbol{z}} D_{\mathsf{KL}}(\mathbb{Q}(. \mid \boldsymbol{z}(0) = \boldsymbol{z}_0) \parallel \widehat{\mathbb{Q}}(. \mid \widehat{\boldsymbol{z}}(0) = \boldsymbol{z}_0))\mathbb{Q}_{\boldsymbol{z}(0)=\boldsymbol{z}_0}(\mathrm{d}\boldsymbol{z}).
$$

From the non-negativity of KL divergence, we obtain the desired results. $\square$

**Lemma 3.** *(Chain Rule of KL Divergence.)*

*Let $\mathbb{Q}$ and $\widehat{\mathbb{Q}}$ be the path measure induced by the two following reverse-time SDEs*

$$
\mathrm{d}\boldsymbol{z} = \nabla \log q_{\sigma_t}(\boldsymbol{z})\mathrm{d}t + \sqrt{2\mathcal{T}_t}\mathrm{d}\boldsymbol{w}_t \quad \boldsymbol{z}(T) = \boldsymbol{z}_T \sim \mathcal{N}(\boldsymbol{0}, \boldsymbol{I})
$$

$$
\mathrm{d}\widehat{\boldsymbol{z}} = \nabla \log \widehat{q}_{\sigma_t}(\widehat{\boldsymbol{z}})\mathrm{d}t + \sqrt{2\mathcal{T}_t}\mathrm{d}\boldsymbol{w}_t \quad \widehat{\boldsymbol{z}}(T) = \widehat{\boldsymbol{z}}_T \sim \mathcal{N}(\boldsymbol{0}, \boldsymbol{I}).
$$

*From the chain rule of KL divergence, we have*

$$
D_{\mathsf{KL}}(\mathbb{Q} \parallel \widehat{\mathbb{Q}}) = D_{\mathsf{KL}}(\mathbb{Q}_{\boldsymbol{z}(T)=\boldsymbol{z}_T} \parallel \widehat{\mathbb{Q}}_{\widehat{\boldsymbol{z}}(T)=\widehat{\boldsymbol{z}}_T})
$$

$$
+ \int_{\boldsymbol{z}} D_{\mathsf{KL}}(\mathbb{Q}(.|\boldsymbol{z}(T) = \boldsymbol{z}_T) \parallel \widehat{\mathbb{Q}}(.|\widehat{\boldsymbol{z}}(T) = \widehat{\boldsymbol{z}}_T))\,\mathbb{Q}_{\boldsymbol{z}(T)=\boldsymbol{z}_T}(\mathrm{d}\boldsymbol{z})
$$

$$
= D_{\mathsf{KL}}(\mathbb{Q}_{\boldsymbol{z}(T)=\boldsymbol{z}_T} \parallel \widehat{\mathbb{Q}}_{\widehat{\boldsymbol{z}}(T)=\widehat{\boldsymbol{z}}_T}) + \mathbb{E}_{\boldsymbol{z}_T \sim \mathcal{N}(\boldsymbol{0}, \boldsymbol{I})}\left[D_{\mathsf{KL}}(\mathbb{Q}(.|\boldsymbol{z} = \boldsymbol{z}_T) \parallel \widehat{\mathbb{Q}}(.|\widehat{\boldsymbol{z}} = \boldsymbol{z}_T))\right]
$$

$$
= \mathbb{E}_{\boldsymbol{z}_T \sim \mathcal{N}(\boldsymbol{0}, \boldsymbol{I})}\left[D_{\mathsf{KL}}(\mathbb{Q}(.|\boldsymbol{z} = \boldsymbol{z}_T) \parallel \widehat{\mathbb{Q}}(.|\widehat{\boldsymbol{z}} = \boldsymbol{z}_T))\right],
$$

*where in the last two equalities, we use the fact that $\mathbb{Q}_{\boldsymbol{z}_T} = \widehat{\mathbb{Q}}_{\boldsymbol{z}_T} = \mathcal{N}(\boldsymbol{0}, \boldsymbol{I})$.*

## B PROOF OF SYMMETRY PROPERTY OF THE MSM SCORE JACOBIAN

We now show that the MSM score has a symmetric Jacobian, as in the true score. Assuming the underlying distribution is twice differentiable and well-behaved (Song & Ermon, 2019; Song et al., 2021b), differentiating the MSM score expression in (3) with respect to $\boldsymbol{z}_t$ gives

$$
\nabla^2 \log q_{\sigma_t}(\boldsymbol{z}_t) = \boldsymbol{W}\,\mathbb{E}_{\boldsymbol{S} \sim p(\boldsymbol{S})}\left[\boldsymbol{S}^\top \nabla^2 \log p_{\sigma_t}(\boldsymbol{s}_t \mid \boldsymbol{S})\,\boldsymbol{S}\Big|_{\boldsymbol{s}_t = \boldsymbol{S}\boldsymbol{z}_t}\right]. \tag{12}
$$

Since $\nabla^2 \log p_{\sigma_t}(s_t \mid S)$ is symmetric by assumption, and both the transformation $S^\top (\cdot) S$ and expectation preserve symmetry, the result remains symmetric; multiplying by the diagonal matrix $W$ also maintains symmetry, so the MSM score admits a symmetric Jacobian.

## C    PRACTICAL RELEVANCE OF ASSUMPTIONS AND THEORY

### C.1    PRACTICAL VALIDITY OF OUR ASSUMPTIONS ON FORWARD-OPERATOR

Our theory assumes two conditions: (i) the training measurements collectively cover the full measurement domain, and (ii) the subsampling operators $S$ used during image sampling are drawn independently from a distribution $p(S)$. Both conditions are realistic in many imaging modalities. For (i), different acquisitions naturally provide complementary subsets of measurements. For example, in magnetic resonance imaging (MRI), different k-space sampling masks are routinely used across acquisitions; in computed tomography (CT), projection angles can be varied; in optical tomography, source–detector positions or wavelengths can change; in positron emission tomography (PET), detector configurations or energy windows may differ; in ultrasound imaging, one can vary transducer firing patterns; and in light-field photography or electron microscopy, system parameters such as aperture, focal length, sample orientation, or defocus levels can be adjusted. Beyond scientific imaging, image-restoration settings also satisfy this condition when different samples reveal different observable regions—for instance, box inpainting, random (dust-like) inpainting, or partially blurred regions that vary across images. MSM can accommodate such scenarios because the observable regions collectively cover the full domain. However, degradations that apply uniformly across the entire image—such as global Gaussian blur, motion blur, or nonlinear measurement effects—do not expose complementary regions and therefore are not handled by the current formulation. Extending MSM to handle such globally applied degradations remains an interesting direction for future work.

For (ii), once the set of forward operators observed in the training data defines the empirical distribution $p(S)$, MSM simply samples new operators independently from this distribution during generation. Note that this assumption is only required during sampling and does not require the physical hardware to acquire measurements independently; it simply reflects how MSM simulates operator variability during generation. Importantly, sampling new operators in this way also ensures that all measurement coordinates are repeatedly visited throughout the diffusion process. If each stochastic loop drops a proportion $p$ of coordinates, then the probability that a given coordinate is never selected across all diffusion steps is $p^{(\#\text{steps} \times w)}$, which becomes negligibly small in practice. For instance, with $p = 0.4$, 100 diffusion steps, and $w = 1$, this probability is $0.4^{100} \approx 1.6 \times 10^{-40}$. Thus, MSM effectively achieves full coverage of the measurement domain during sampling, avoiding the undesirable case where any region is left untouched.

### C.2    EMPIRICAL ACCESSIBILITY OF BOUND PARAMETERS IN THEORY

Theorem 1 establishes that the KL divergence between the true distribution $q(z)$ and its stochastic approximation $\widehat{q}(z)$ is bounded as

$$D_{\mathsf{KL}}(q(z) \parallel \widehat{q}(z)) \leq \frac{v^2}{w} C, \tag{13}$$

where $v$ quantifies the variance of the stochastic score approximation and $C$ is a finite constant determined by the diffusion process.

To estimate the range of $v$ in practice, we computed the squared error between the MSM score, approximated using $w = 64$ mini-batches, and its stochastic approximation:

$$\sum_{i=1}^{n} \left\| \nabla \log q_{\sigma_t}\left( z^{(i)} \right) - \nabla \log \widehat{q}_{\sigma_t}\left( z^{(i)} \right) \right\|_2^2, \tag{14}$$

averaged over 3,000 training images perturbed at timesteps $t = 200, 400, 600$. By varying the number of mini-batches $w \in \{1, 4, 16\}$, we directly observed that the approximation error with MSM score (which is approximated with $w = 64$) decreases with $w$, consistent with the theoretical scaling $v^2/w$ as in Table X for both FFHQ and fastMRI datasets.

Table 5: Empirical squared error between the MSM score (reference $w = 64$) and its stochastic approximation for different timesteps $t$ and minibatch sizes $w$. Results are averaged over 3,000 training images for FFHQ and fastMRI.

| Time step $t$ | #Minibatches $w$ | $\sum_{i=1}^{n} \|\nabla \log q_{\sigma_t}(\boldsymbol{z}^{(i)}) - \nabla \log \widehat{q}_{\sigma_t}(\boldsymbol{z}^{(i)})\|_2^2$ | |
|---|---|---|---|
| | | FFHQ | fastMRI |
| | 16 | $3.57 \times 10^{-3}$ | $4.15 \times 10^{-5}$ |
| 200 | 4 | $5.80 \times 10^{-2}$ | $1.15 \times 10^{-4}$ |
| | 1 | $6.17 \times 10^{-1}$ | $4.05 \times 10^{-4}$ |
| | 16 | $8.20 \times 10^{-4}$ | $4.65 \times 10^{-6}$ |
| 400 | 4 | $5.70 \times 10^{-3}$ | $1.12 \times 10^{-5}$ |
| | 1 | $3.20 \times 10^{-2}$ | $2.79 \times 10^{-5}$ |
| | 16 | $5.60 \times 10^{-4}$ | $1.42 \times 10^{-6}$ |
| 600 | 4 | $2.70 \times 10^{-3}$ | $4.17 \times 10^{-6}$ |
| | 1 | $3.70 \times 10^{-3}$ | $6.21 \times 10^{-6}$ |

For $C$, we recall from the proof of Theorem 1 that

$$C = \int_0^T \frac{1}{4\mathcal{T}_t} \, dt, \tag{15}$$

where $\mathcal{T}_t$ is the temperature associated with the variance-preserving diffusion in our experiment setup. Under this setting, $\mathcal{T}_t = \frac{1}{2}(1 - \alpha_t)/\alpha_t$ in terms of the signal scaling factor $\alpha_t$ that defines the noisy image at diffusion step $t$, as specified in (Park et al., 2025, Table 1). Hence,

$$C = \int_0^T \frac{1 - \alpha_t}{4\alpha_t} \, dt. \tag{16}$$

Using the schedule parameters from prior work, we obtain $C = 101.01$ for our unconditional MSM setup.

These results demonstrate that both $v$ and $C$ are not abstract constants but measurable quantities. In particular, $v$ can be empirically estimated from score approximation error, and $C$ admits a closed-form expression based on the diffusion temperature schedule.

## D    IMPLEMENTATION DETAILS

### D.1    MSM FRAMEWORK IN MR IMAGES

The main manuscript illustrates the measurement score-based diffusion model *(MSM)* framework's training and sampling schemes, but omits domain-specific transformations for clarity. These transformations are essential in the MRI setting, which requires conversions between measurement and image spaces before and after denoising.

Specifically, we apply the inverse Fourier transform $\boldsymbol{F}^\top$ followed by the adjoint coil-sensitivity operator $\boldsymbol{C}^\top$ to project the measurements into image space before denoising. After denoising, we map the denoised image back to measurement space by applying the forward coil-sensitivity operator $\boldsymbol{C}$ and the Fourier transform $\boldsymbol{F}$.

This results in a modified version of (1), given by:

$$\hat{s}_\theta(\boldsymbol{s}_t \,;\, \sigma_t) \leftarrow \boldsymbol{SFC}\,\mathsf{D}_\theta(\boldsymbol{C}^\top \boldsymbol{F}^\top \boldsymbol{s}_t \,;\, \sigma_t), \tag{17}$$

where $\boldsymbol{S}$ is the subsampling operator. A visual illustration is provided in Figure 4.

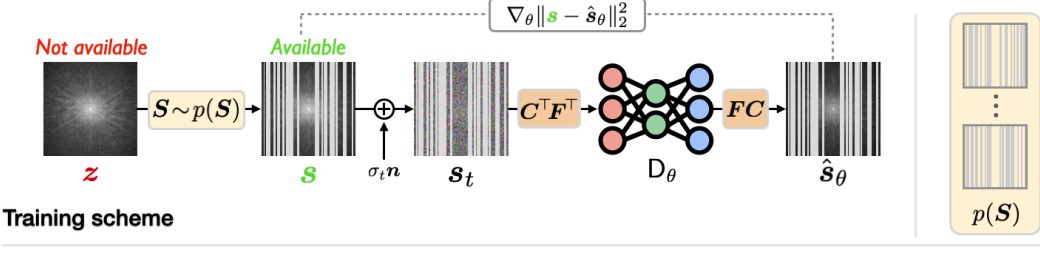

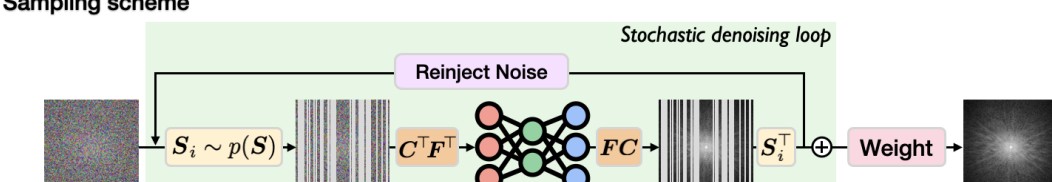

Figure 4: Illustration of the *Measurement Score-based diffusion Model (MSM)* for training and sampling using subsampled MRI measurements. MSM operates directly on k-space measurements, with minor domain transformations between k-space and image space only at the input and output of the diffusion model $\mathsf{D}_\theta$.

### D.2    DETAILED MSM POSTERIOR SAMPLING ALGORITHM

This appendix provides the dedicated algorithm for the MSM-based posterior sampling procedure introduced in Section 3.3. Algorithm 2 extends the unconditional MSM sampler by inserting, at each diffusion step, a data-consistency update derived from the posterior gradient. This makes explicit how MSM transitions from unconditional sampling to solving inverse problems.

### D.3    STOCHASTIC POSTERIOR SAMPLING FOR COMPRESSED-SENSING MRI

Our posterior sampling algorithm is described in Section 3.3. For compressed-sensing MRI, we apply an additional simplification based on directly approximating the *partial posterior score* for each partially subsampled measurement within the stochastic algorithm.

---

**Algorithm 2** Measurement Score-Based Posterior Sampling

---

**Require:** $T, p(\boldsymbol{S}), \{\sigma_t\}_{t=1}^T, \boldsymbol{y}, \boldsymbol{H}$

1: Initialize $\boldsymbol{z}_T \sim \mathcal{N}(\boldsymbol{0}, \boldsymbol{I})$, $\hat{\boldsymbol{z}}_\theta \leftarrow \boldsymbol{0}$

2: **for** $t = T$ **to** 1 **do**

3:     **for** $i = 1$ **to** $w$ **do**

4:         $\boldsymbol{S}^{(i)} \sim p(\boldsymbol{S})$,   $\boldsymbol{s}_t^{(i)} \leftarrow \boldsymbol{S}^{(i)} \boldsymbol{z}_t$                           *Partial score-based denoising*

5:         $\hat{\boldsymbol{s}}_\theta^{(i)} \leftarrow \boldsymbol{s}_t^{(i)} + \sigma_t^2 \mathsf{S}_\theta(\boldsymbol{s}_t^{(i)}; \sigma_t, \boldsymbol{S}^{(i)})$

6:         $\boldsymbol{s}_t^{(i)} \sim p(\boldsymbol{s}_t^{(i)} \mid \hat{\boldsymbol{s}}_\theta^{(i)})$                                 *Reinject noise & update iterate*

7:         $\boldsymbol{z}_t \leftarrow \boldsymbol{S}^{(i)\top} \boldsymbol{s}_t^{(i)} + (\boldsymbol{I} - \boldsymbol{S}^{(i)\top} \boldsymbol{S}^{(i)}) \boldsymbol{z}_t$

8:     **end for**

9:     $\boldsymbol{C} \leftarrow \sum_{i=1}^w \mathrm{diag}(\boldsymbol{S}^{(i)\top} \boldsymbol{S}^{(i)})$                      *Compute weight for aggregation*

10:    $\boldsymbol{W} \leftarrow [\max(\boldsymbol{C}, 1)]^{-1}$

11:    $\hat{\boldsymbol{z}}_\theta \leftarrow \boldsymbol{W} \sum_{i=1}^w \boldsymbol{S}^{(i)\top} \hat{\boldsymbol{s}}_\theta^{(i)} + \mathbb{1}_{C=0} \cdot \hat{\boldsymbol{z}}_\theta$        *Aggregate partial denoised results*

12:    $\hat{\boldsymbol{z}}_\theta \leftarrow \hat{\boldsymbol{z}}_\theta - \gamma_t \nabla_{\hat{\boldsymbol{z}}_\theta} \|\boldsymbol{y} - \boldsymbol{H} \hat{\boldsymbol{z}}_\theta\|_2^2$

13:    $\boldsymbol{z}_{t-1} \sim p(\boldsymbol{z}_{t-1} \mid \boldsymbol{z}_t, \hat{\boldsymbol{z}}_\theta)$

14: **end for**

15: **return** $\boldsymbol{z}_0$

---

To perform posterior sampling for compressed-sensing MRI, we estimate the posterior score using a stochastic ensemble, similar to the MSM score ensemble in Section 3.2:

$$
\begin{aligned}
\nabla \log p_{\sigma_t}(\boldsymbol{z}_t \mid \boldsymbol{y}) &= \nabla \log p_{\sigma_t}(\boldsymbol{z}_t) + \nabla \log p_{\sigma_t}(\boldsymbol{y} \mid \boldsymbol{z}_t) \\
&\approx \boldsymbol{W} \left[ \frac{1}{w} \sum_{i=1}^w \boldsymbol{S}^{(i)\top} \nabla \log p_{\sigma_t}(\boldsymbol{s}_t^{(i)} \mid \boldsymbol{y}^{(i)}) \Big|_{\boldsymbol{s}_t^{(i)} = \boldsymbol{S}^{(i)} \boldsymbol{z}_t} \right] \\
&= \boldsymbol{W} \left[ \frac{1}{w} \sum_{i=1}^w \boldsymbol{S}^{(i)\top} \left( \nabla \log p_{\sigma_t}(\boldsymbol{s}_t^{(i)}) + \nabla \log p_{\sigma_t}(\boldsymbol{y}^{(i)} \mid \boldsymbol{s}_t^{(i)}) \right) \Big|_{\boldsymbol{s}_t^{(i)} = \boldsymbol{S}^{(i)} \boldsymbol{z}_t} \right],
\end{aligned}
\tag{18}
$$

where the second line replaces the partial prior score in the stochastic MSM approximation (5) with the corresponding partial posterior score, and the third line expands this term using Bayes' rule into prior and likelihood components. All partial prior and likelihood distributions are conditioned on the corresponding subsampling operator $\boldsymbol{S}^{(i)}$; for notational simplicity, we omit this conditioning in $p_{\sigma_t}(\boldsymbol{s}_t^{(i)})$ and $p_{\sigma_t}(\boldsymbol{y}^{(i)} \mid \boldsymbol{s}_t^{(i)})$. For each subsampling operator $\boldsymbol{S}^{(i)} \in \mathbb{R}^{m_i \times n}$, we define $\boldsymbol{y}^{(i)} = \boldsymbol{S}^{(i)} \boldsymbol{H}^\top \boldsymbol{y}$ to project the observed measurement $\boldsymbol{y}$ into the same partial measurement space as $\boldsymbol{s}_t^{(i)}$. The log-likelihood gradient is approximated by

$$
\begin{aligned}
\nabla \log p_{\sigma_t}(\boldsymbol{y}^{(i)} \mid \boldsymbol{s}_t^{(i)}) &\approx \nabla \log p_{\sigma_t}(\boldsymbol{y}^{(i)} \mid \hat{\boldsymbol{s}}_\theta^{(i)}) \\
&= \gamma_t \nabla \left\| \boldsymbol{y}^{(i)} - \tilde{\boldsymbol{H}}^{(i)} \hat{\boldsymbol{s}}_\theta^{(i)} \right\|_2^2,
\end{aligned}
\tag{19}
$$

where $\gamma_t$ is a tunable step size, and we define $\tilde{\boldsymbol{H}}^{(i)} = \boldsymbol{S}^{(i)} \boldsymbol{H}^\top \boldsymbol{H} \boldsymbol{S}^{(i)\top}$ as the degradation operator $\boldsymbol{H}^\top \boldsymbol{H}$ restricted to the coordinates selected by $\boldsymbol{S}^{(i)}$.

### D.4 Model Architecture and Training Configuration

We adopted the U-Net architecture (Ronneberger et al., 2015), following the design used in (Ho et al., 2020; Dhariwal & Nichol, 2021), as our diffusion model backbone. Models were trained with the AdamW optimizer (Loshchilov & Hutter, 2019) and used an exponential moving average

(EMA) to stabilize training by averaging model weights over time, using a decay rate of 0.9999 for gradual updates. The diffusion process consisted of 1000 timesteps, with a linearly increasing noise variance schedule starting from 0.0001 and reaching 0.2 at the final step. All diffusion models, including diffusion-based baselines, were trained with the same architecture for each application. The hyperparameter setup and architectural details are summarized in Table 6.

Table 6: Diffusion model architecture and training hyperparameters for each dataset.

|  | RGB Face Images | Multi-Coil MRI |
|---|---|---|
| **Base channel width** | 128 | |
| **Attention resolutions** | [32, 16, 8] | |
| **# Attention heads** | 4 | |
| **# Residual blocks** | 2 | |
| **Batch size** | 128 | 32 |
| **Learning rate** | $5e-5$ | $1e-5$ |
| **Channel multipliers** | [1, 1, 2, 3, 4] | [1, 1, 2, 2, 4, 4] |
| **# Input/Output channels** | 3 | 2 |

### D.5 COMPARISON METHODS FOR UNCONDITIONAL SAMPLING

We now provide detailed setups for used baselines for unconditional sampling experiments.

**Oracle diffusion.** For both natural images and multi-coil MRI, we train the oracle diffusion model using clean images without any degradation. The model is trained to predict the noise component of noisy images for both data types.

**Ambient diffusion (Daras et al., 2024c).** For natural images, under the same training setup as MSM in noiseless and subsampled data scenario, following the recommendation of (Daras et al., 2024c) to apply minimal additional corruption, we define the further degradation operator $\tilde{S}$ by dropping one additional $32 \times 32$ pixel box. The model is trained to directly predict the clean image, which we found to perform better than predicting the noise component.

For MRI data, under the same training setup as the noiseless and subsampled setting of MSM, to define the further degradation operator $\tilde{S}$, we drop an additional $10\%$ of the sampling pattern while preserving the autocalibration signal region. Unlike the natural image case, the model is trained to predict the noise component, which we found to perform better than direct clean image prediction.

During sampling, we use 200 steps of denoising diffusion implicit models (DDIM) (Song et al., 2021a) and apply the same further degradation configuration used during training to subsample the diffusion iterate.

**GSURE diffusion (Kawar et al., 2024).** We only apply this method to the RGB face data, not the multi-coil MRI data, because GSURE diffusion's extension from single-coil MRI to multi-coil MRI remains computationally infeasible, and no practical approach has been proposed. Under the same training setup of MSM's noisy and subsampled training setup, we train the GSURE diffusion model to predict the clean images and follow exactly the same training configuration as described in (Kawar et al., 2024).

Note that we exclude recent expectation-maximization-based methods (Bai et al., 2024; 2025), as their reliance on clean-image initialization is incompatible with our setting, where no clean images are available.

### D.6 COMPARISON METHODS FOR IMAGING INVERSE PROBLEMS

We now describe the baseline methods used for solving inverse problems.

**Diffusion posterior sampling (DPS) (Chung et al., 2023).** DPS estimates the gradient of the log-likelihood using the MMSE estimate $\hat{x}_\theta(x_t)$ from a pretrained diffusion model as

$$\nabla \log p(\boldsymbol{y} \mid \boldsymbol{x}_t) \approx \nabla \log p(\boldsymbol{y} \mid \hat{\boldsymbol{x}}_\theta(\boldsymbol{x}_t)), \tag{20}$$

where the gradient is taken with respect to $x_t$.

Following the original implementation, we set the step size for the likelihood gradient as $\gamma_t = \frac{c}{\|y - A\mathbb{E}[x_0|x_t]\|_2}$, where $c$ is selected via grid search within the recommended range $[0.1, 10]$. We used $c = 2$ for the super-resolution experiment, $c = 0.7$ for box inpainting, and $c = 10$ for compressed sensing MRI.

**Ambient diffusion posterior sampling (A-DPS) (Aali et al., 2025).** A-DPS follows the same posterior sampling strategy as DPS but uses a diffusion model trained on noiseless subsampled data. For natural-image experiments, we use the pretrained Ambient diffusion model trained with a dropping ratio $p = 0.4$. For MRI experiments, we use the pretrained Ambient diffusion model trained at acceleration rate $R = 4$, consistent with the standard setup in diffusion-based MRI reconstruction. The step size is set in the same form as DPS: $\gamma_t = \frac{c}{\|y - A\mathbb{E}[x_0|x_t]\|_2}$, with the constant $c$ chosen according to each inverse problem: $c = 2$ for super-resolution, $c = 0.7$ for box inpainting, and $c = 10$ for compressed sensing MRI.

**Robust self-supervision via data undersampling (Robust SSDU) (Millard & Chiew, 2024).** Robust SSDU is designed to handle noisy, subsampled measurements by introducing additional sub-sampling and noise to the observed measurements during training. In our implementation, the primary sampling mask $S_1$ exactly matches the test acquisition patterns (acceleration factors $\{2, 4, 6, 8\}$), and the training noise level is set to the same value used at test time ($\sigma_n \in \{0.005, 0.01, 0.02, 0.03\}$). Given the noisy subsampled input $s = S_1 z + n$, where $n$ is additive Gaussian noise with standard deviation $\sigma_n$, Robust SSDU forms a further corrupted input

$$\tilde{s} = S_2 s + \tilde{n}, \tag{21}$$

where $S_2$ has an acceleration factor of $R = 2$, and $\tilde{n}$ is independent Gaussian noise with standard deviation $\sigma_n$, matching the original measurement noise level, following the original implementation.

**Denoising diffusion null-space model (DDNM) (Wang et al., 2023a).** DDNM also uses a diffusion model trained on clean data and introduces a projection-based update that blends the prior estimate and the measurement:

$$\mathbb{E}[x_0 \mid x_t, y] \approx (I - \Sigma_t A^\dagger A)\mathbb{E}[x_0 \mid x_t] + \Sigma_t A^\dagger y, \tag{22}$$

where $A^\dagger$ is the pseudoinverse and $\Sigma_t$ is a weighting matrix, such as $\Sigma_t = \lambda_t I$ or a spectrally tuned version.

We followed the enhanced version of DDNM described in (Wang et al., 2023a, Section 3.3 and Equation (19)) to specify weight matrix $\Sigma_t$ in (22).

### D.7 MEASURING FID SCORE

To compute the Fréchet Inception Distance (FID), we used the implementation provided in the following repository[1]. For each method, we generated 10,000 images, then computed FID using features extracted from a pretrained inception network. For MRI images, which have complex-valued channels, we converted them to magnitude images and replicated the single-channel magnitude three times to form a 3-channel input compatible with the pretrained Inception network. Note that although the pretrained FID model was trained on natural images, it still reflects perceptual quality on MRI data (Bendel et al., 2023).

---

[1] https://github.com/mseitzer/pytorch-fid

### D.8 Illustration of Subsampled Measurements and Their Scores

The main manuscript establishes that MSM learns the score of each subsampled measurement—implicitly conditioned on the corresponding subsampling mask—and shows how these measurement scores are leveraged to synthesize fully-sampled measurements. The purpose of this section is to visually illustrate how these measurement scores behave under different masks and to clarify the dimensional structure underlying MSM sampling.

Recall that MSM operates on a subsampled measurement $s_t \in \mathbb{R}^m$ formed as $s_t = S\,z_t$, where $z_t \in \mathbb{R}^n$ is not accessible fully-sampled measurement and $S \in \{0,1\}^{m \times n}$ with $m < n$ is a binary subsampling mask. Because each $s_t$ is produced by a specific mask $S$, the measurement score $\nabla \log p_{\sigma_t}(s_t)$ is implicitly mask-conditioned: it describes the distribution of $s_t$ restricted to the coordinates selected by that mask, rather than a score marginalized over all masks.

Figure 5 provides a concrete visual example using a $64 \times 64$ image. The figure illustrates: (i) how different masks $S$ produce different subsampled measurements $s_t$, (ii) how the dimensionality changes from the fully-sampled measurement to each subsampled measurement; and (iii) how MSM performs score-based denoising under several predefined masks. This visualization makes explicit how the measurement scores are inherently mask-conditioned and how MSM integrates them during sampling to reconstruct the fully-sampled measurement.

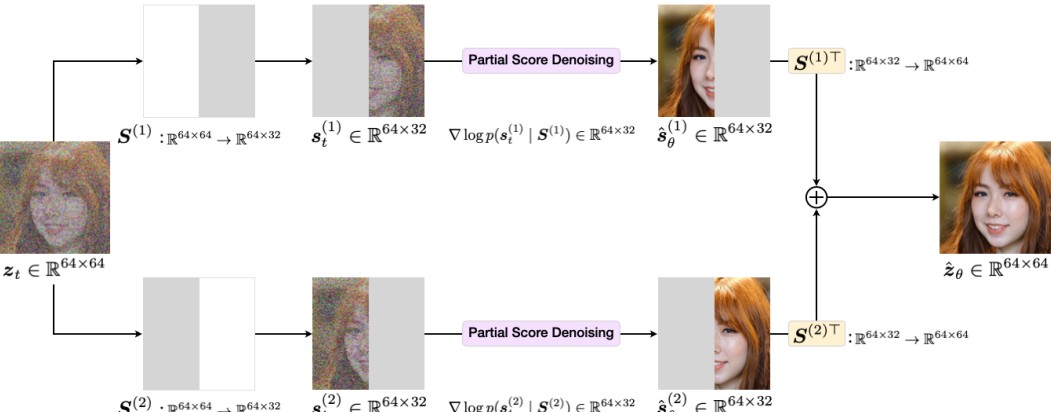

Figure 5: Visual illustration of subsampled measurements, their associated masks, and the corresponding measurement scores used in MSM sampling.

# E  ABLATION STUDIES AND DISCUSSIONS

## E.1  PRIOR WORK AND DISTINCTION OF OUR APPROACH

Among recent approaches that train diffusion models without clean images, ambient diffusion (Daras et al., 2024c) and its posterior-sampling extension A-DPS (Aali et al., 2025) are the most directly related, as they also learn from degraded natural images and multi-coil MRI. However, both operate in the *image domain*, approximating clean-image scores by further subsampling images and predicting the original subsampled images. In contrast, MSM generalizes the idea of patch-based learning—widely applied to enable computationally efficient supervised training—to self-supervised learning in the measurement domain. It learns partial *measurement scores* restricted to observable regions from only noisy, subsampled measurements, and combines them in expectation to form a full measurement score. With our efficient stochastic algorithm, MSM enables both unconditional sampling and posterior sampling for solving linear inverse problems. Table 7 highlights the key differences.

Table 7: Comparison of MSM with ambient diffusion (Daras et al., 2024c) and its direct extension Ambient Diffusion Posterior Sampling (A-DPS) (Aali et al., 2025). MSM is the first to directly learn measurement scores from noisy, subsampled measurements and to use them for both generating full measurements and solving inverse problems.

| | **Ambient Diffusion** | **Ambient-DPS** | **MSM (Ours)** |
|---|---|---|---|
| **Training data** | Subsampled data, with further random subsampling in training | | Only noisy, subsampled measurements |
| **Learned object** | Learns the approximation of the true score by predicting the original subsampled image from a further subsampled input | | *Partial measurement scores*, restricted to observable regions |
| **Sampling domain** | Image domain | | Measurement domain: stochastic generation of the full measurements |
| **Posterior sampling** | Limited: can only solve inverse problems when the test-time degradation matches the training degradation | Direct extension of diffusion posterior sampling (Chung et al., 2023) applied to pretrained ambient diffusion | New posterior score formulation using partial measurement posterior scores; stochastic conditional sampling of full measurements |
| **Key novelty** | First diffusion framework trained purely on subsampled data | Extension of ambient diffusion to posterior sampling | First to *directly learn measurement scores* and use them in a stochastic measurement-space diffusion process for both full measurement and posterior sampling |

### E.2 IMAGE SAMPLING WITH MSM USING EXTREMELY SUBSAMPLED DATA

We evaluated the unconditional sampling capability of our framework in a more challenging training scenario with extremely subsampled data. Specifically, we used MRI data with k-space subsampling via random masks at an acceleration factor of $R = 8$, including fully-sampled vertical lines and the central 20 lines for autocalibration.

MSM was configured with a stochastic loop parameter $w = 2$ and took 200 sampling steps. As a baseline, ambient diffusion took 200 sampling steps. As shown in Table 8 and Figure 6, MSM achieves a lower FID score than the Ambient diffusion on the same data setup, demonstrating the robustness of our approach under high subsampling rates.

Table 8: FID scores under different training settings on multi-coil brain MR images. **Best values** are highlighted for each training scenario, with comparisons shown when corresponding baseline methods are available. Note how MSM consistently achieves lower FID scores than the Ambient diffusion, even in extremely subsampled data scenarios.

| Training data | Methods | FID↓ |
|---|---|---|
| No degradation | Oracle diffusion | 29.25 |
| $R = 4, \rho = 0$ | MSM | **43.60** |
| | Ambient diffusion | 47.80 |
| $R = 8, \rho = 0$ | MSM | **74.92** |
| | Ambient diffusion | 84.77 |

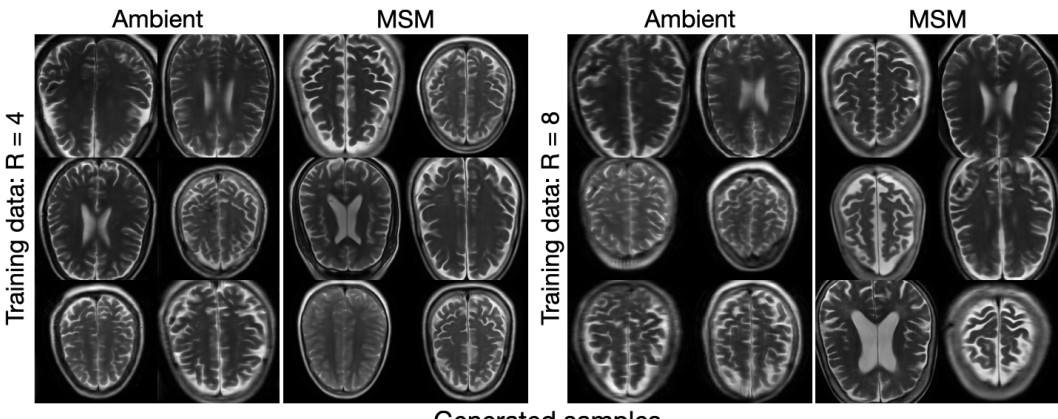

Figure 6: Visual comparison of MSM trained under extreme subsampling ($R = 8$) with MSM and baseline methods trained under less degraded conditions.

### E.3 SOLVING INVERSE PROBLEMS WITH MSM TRAINED ON NOISY AND SUBSAMPLED DATA

We showed that MSM trained on subsampled data can solve inverse problems for both natural images and multi-coil MRI. We further verified that MSM, when trained on noisy and subsampled data, can achieve comparable performance using the same step size for the log-likelihood gradient as in the subsampled-only scenario, as summarized in Table 9 and Figure 7.

Table 9: Results on two natural image inverse problems and two compressed sensing MRI tasks. MSM trained on subsampled data is compared with MSM trained on noisy and subsampled data (training type shown in parentheses). Both training setups yield comparable performance across all metrics.

| Testing data | | Input | MSM (Noisy & Subsampled) | MSM (Noiseless & Subsampled) |
|---|---|---|---|---|
| **Box Inpainting** | PSNR↑ | 18.26 | 24.16 | 24.71 |
| | SSIM↑ | 0.749 | 0.864 | 0.867 |
| | LPIPS↓ | 0.304 | 0.081 | 0.076 |
| **SR** (×4) | PSNR↑ | 23.21 | 27.99 | 28.11 |
| | SSIM↑ | 0.728 | 0.868 | 0.868 |
| | LPIPS↓ | 0.459 | 0.127 | 0.117 |
| **CS-MRI** (×4) | PSNR↑ | 22.75 | 29.74 | 30.71 |
| | SSIM↑ | 0.648 | 0.826 | 0.839 |
| | LPIPS↓ | 0.306 | 0.168 | 0.145 |
| **CS-MRI** (×6) | PSNR↑ | 21.94 | 28.11 | 28.86 |
| | SSIM↑ | 0.617 | 0.795 | 0.805 |
| | LPIPS↓ | 0.342 | 0.192 | 0.168 |

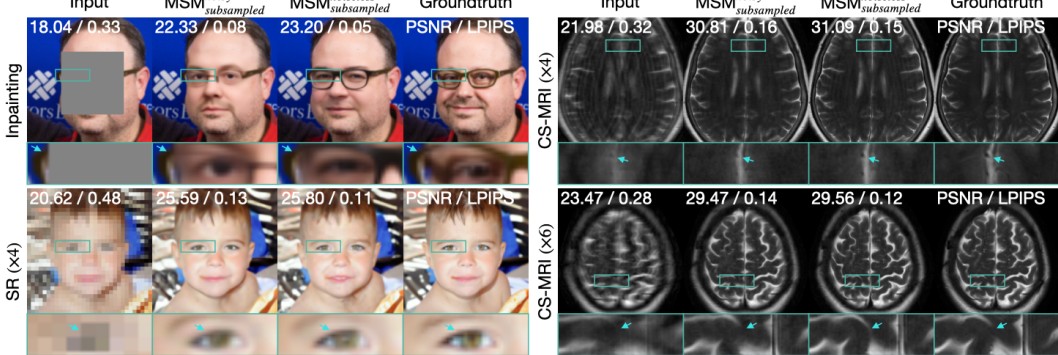

Figure 7: Visual comparison between MSM trained on noisy and subsampled data and MSM trained on only subsampled data. Both models produce high-quality results on natural image and MRI tasks.

### E.4 COMPARISON WITH CLASSICAL CS-MRI RECONSTRUCTION METHOD

We have shown that MSM outperforms diffusion-based and self-supervised methods in multi-coil compressed sensing MRI, all trained without clean data. We now compare MSM with the classical CS-MRI method: total variation (TV) regularization.

Table 10: Quantitative results on two compressed sensing MRI tasks. MSM, trained using only subsampled data, is compared with the classical CS-MRI method: total variation (TV) regularization.

| Setup | | Input | TV | MSM |
|---|---|---|---|---|
| **CS-MRI** (×4) | PSNR↑ | 22.75 | 25.71 | **30.71** |
| | SSIM↑ | 0.648 | 0.750 | **0.839** |
| | LPIPS↓ | 0.306 | 0.238 | **0.145** |
| **CS-MRI** (×6) | PSNR↑ | 21.94 | 24.18 | **28.86** |
| | SSIM↑ | 0.617 | 0.702 | **0.805** |
| | LPIPS↓ | 0.342 | 0.282 | **0.168** |

### E.5 COMPARISON WITH DIFFUSION-BASED INVERSE PROBLEM SOLVERS TRAINED ON CLEAN DATA

We compared MSM against diffusion-based inverse problem solvers—DPS (Chung et al., 2023) and DDNM (Wang et al., 2023a)—that use diffusion priors trained on clean images, on both RGB face images and multi-coil compressed sensing MRI. Detailed configurations of these methods are provided in Section D.6.

The measurement noise level was set to $\eta = 0.01$, and all experimental setups for both MSM and the baselines followed those described in Section 4.1 and Section 4.2. The results are summarized in Table 11 and Figure 8. Notably, although MSM was trained only on subsampled measurements, it performs comparably to the baselines that leverage clean data-based pretrained diffusion models and even surpasses DPS on several inverse problems across multiple metrics. This result highlights that reconstruction quality depends not only on the diffusion prior but also on the data consistency strategy. Because DDNM employs a data consistency strategy similar to ours, it achieves slightly better results than ours; in contrast, DPS relies on a different data consistency strategy, which leads to its distinct and generally lower performance even when using the clean diffusion prior.

Table 11: Quantitative results on two natural image inverse problems and two compressed sensing MRI tasks. MSM, trained using only subsampled data, is compared with methods that use pretrained diffusion models trained on clean data. The number of iterations used for each method is shown in parentheses. **Best** and second-best values are highlighted per metric (PSNR, SSIM, LPIPS). Note that despite not having access to the clean data, MSM approaches the performance of the clean data-based methods.

| Testing data | | Input | DPS (1000) | DDNM (200) | MSM (200) |
|---|---|---|---|---|---|
| **Box Inpainting** | PSNR↑ | 18.26 | 23.64 | **25.16** | 24.71 |
| | SSIM↑ | 0.749 | 0.864 | **0.883** | 0.867 |
| | LPIPS↓ | 0.304 | 0.077 | **0.071** | 0.076 |
| **SR** (×4) | PSNR↑ | 23.21 | 27.20 | **28.82** | 28.11 |
| | SSIM↑ | 0.728 | 0.841 | **0.897** | 0.868 |
| | LPIPS↓ | 0.459 | 0.128 | 0.126 | **0.117** |
| **CS-MRI** (×4) | PSNR↑ | 22.75 | 31.31 | **32.84** | 30.71 |
| | SSIM↑ | 0.648 | 0.845 | **0.895** | 0.839 |
| | LPIPS↓ | 0.306 | 0.112 | **0.104** | 0.145 |
| **CS-MRI** (×6) | PSNR↑ | 21.94 | 29.19 | **29.95** | 28.86 |
| | SSIM↑ | 0.728 | 0.795 | **0.851** | 0.805 |
| | LPIPS↓ | 0.459 | 0.149 | **0.141** | 0.168 |

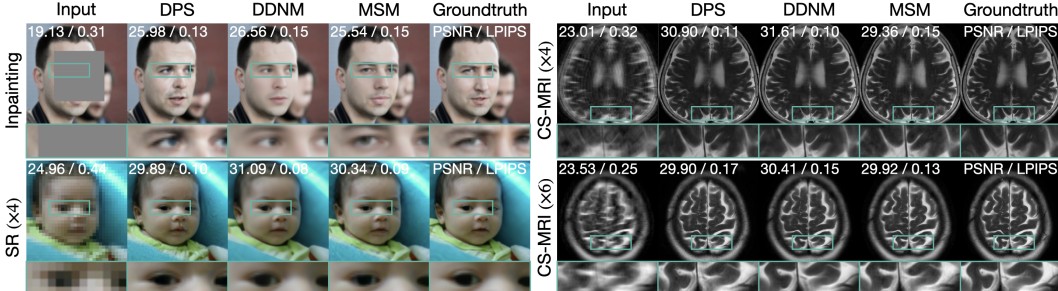

Figure 8: Visual comparison of reconstructed images using diffusion-based inverse problem solvers.

### E.6 Effect of Stochastic Loop Iterations on Sample Quality

Our MSM framework includes a configurable parameter $w$, which controls the number of stochastic iterations used to denoise the full measurement iterates. As theoretically justified in Appendix A, using a larger $w$ yields a closer convergence to the ideal MSM sampling algorithm that averages infinitely many partial scores at each iteration.

To empirically validate this, we fixed the number of diffusion sampling steps to 10 and explored three different values of $w$ in $\{1, 2, 4\}$. We intentionally fixed the low number of sampling steps to isolate the effect of $w$, as increasing the number of diffusion steps improves sample quality. Figure 9 illustrates that larger $w$ leads to visually more plausible generations: while $w = 1$ can still produce reasonable samples, it often results in artifacts, such as visible boundaries in some regions. As $w$ increases, the results become more stable and visually coherent. Table 12 and Table 13 further support these observations quantitatively, showing that larger $w$ achieves better FID scores across both datasets.

Table 12: FID and average time per sampling for MSM with varying stochastic loop iterations $w$ on face images. Note that the trade-off exists where larger $w$ reduces FID but increases sampling time.

| Training data | Methods | FID↓ | Time ($s$) |
|---|---|---|---|
| | MSM ($w = 4$) | 85.02 | 1.02 |
| $p = 0.4, \rho = 0$ | MSM ($w = 2$) | 90.65 | 0.51 |
| | MSM ($w = 1$) | 125.59 | 0.27 |

Table 13: FID and average time per sampling for MSM with varying stochastic loop iterations $w$ on MR images. Similar to face images, increasing $w$ reduces FID at the cost of longer sampling time.

| Training data | Methods | FID↓ | Time ($s$) |
|---|---|---|---|
| | MSM ($w = 4$) | 75.08 | 2.18 |
| $R = 4, \rho = 0$ | MSM ($w = 2$) | 81.97 | 1.12 |
| | MSM ($w = 1$) | 102.89 | 0.57 |

Figure 9: Samples generated using different values of the stochastic loop parameter $w$.

### E.7 Solving Inverse Problems Under Different Test Noise Levels Using MSM

We additionally evaluate how MSM's posterior solver behaves under varying measurement noise at inference time by measuring reconstruction quality across multiple test noise levels while keeping the pretrained measurement-domain diffusion model fixed. As summarized in Table 14, MSM maintains strong performance across all noise levels and generally outperforms competing methods trained without clean images.

Table 14: Reconstruction results across different test noise levels for inpainting, super-resolution, and CS-MRI. **Best values** per metric are highlighted. MSM demonstrates robust performance across noise levels and generally outperforms alternative methods trained without clean images.

| Setup | Noise level | | Input | A-DPS | SSDU | MSM |
|---|---|---|---|---|---|---|
| **Inpainting** | $\eta = 0.005$ | PSNR↑ | 18.29 | 19.26 | N/A | **24.75** |
| | | SSIM↑ | 0.757 | 0.654 | N/A | **0.874** |
| | | LPIPS↓ | 0.299 | 0.288 | N/A | **0.068** |
| **Inpainting** | $\eta = 0.01$ | PSNR↑ | 18.26 | 20.14 | N/A | **24.71** |
| | | SSIM↑ | 0.749 | 0.621 | N/A | **0.867** |
| | | LPIPS↓ | 0.304 | 0.305 | N/A | **0.076** |
| **Inpainting** | $\eta = 0.02$ | PSNR↑ | 18.19 | 19.16 | N/A | **24.68** |
| | | SSIM↑ | 0.726 | 0.657 | N/A | **0.854** |
| | | LPIPS↓ | 0.322 | 0.288 | N/A | **0.089** |
| **SR** (×4) | $\eta = 0.005$ | PSNR↑ | 23.27 | 22.13 | N/A | **28.29** |
| | | SSIM↑ | 0.734 | 0.696 | N/A | **0.880** |
| | | LPIPS↓ | 0.456 | 0.287 | N/A | **0.107** |
| **SR** (×4) | $\eta = 0.01$ | PSNR↑ | 23.21 | 22.61 | N/A | **28.11** |
| | | SSIM↑ | 0.728 | 0.702 | N/A | **0.868** |
| | | LPIPS↓ | 0.459 | 0.277 | N/A | **0.117** |
| **SR** (×4) | $\eta = 0.02$ | PSNR↑ | 23.09 | 22.41 | N/A | **27.59** |
| | | SSIM↑ | 0.711 | 0.702 | N/A | **0.836** |
| | | LPIPS↓ | 0.470 | 0.278 | N/A | **0.147** |
| **CS-MRI** (×4) | $\eta = 0.005$ | PSNR↑ | 22.67 | 27.01 | 29.70 | **31.00** |
| | | SSIM↑ | 0.652 | 0.777 | 0.855 | **0.858** |
| | | LPIPS↓ | 0.305 | 0.197 | 0.165 | **0.127** |
| **CS-MRI** (×4) | $\eta = 0.01$ | PSNR↑ | 22.62 | 27.28 | 29.65 | **30.71** |
| | | SSIM↑ | 0.648 | 0.804 | **0.847** | 0.839 |
| | | LPIPS↓ | 0.306 | 0.173 | 0.160 | **0.145** |
| **CS-MRI** (×4) | $\eta = 0.02$ | PSNR↑ | 22.52 | 27.12 | 28.99 | **29.07** |
| | | SSIM↑ | 0.623 | 0.792 | **0.833** | 0.769 |
| | | LPIPS↓ | 0.329 | 0.178 | **0.168** | 0.205 |

## E.8 Effect of Noise Reinjection in MSM Sampling

As introduced in Section 3.2, MSM includes a noise-reinjection step inside each stochastic loop. At every loop iteration, the diffusion noise is added back to the denoised subsampled estimate before proceeding to the next update. This step helps ensure that coordinates updated in earlier iterations remain compatible with the current iterate. In this section, we compare reconstruction performance with and without this noise-reinjection step.

Table 15: Ablation study on the effect of noise reinjection in MSM sampling. We report quantitative results on natural image inpainting, $4\times$ super-resolution, and CS-MRI at acceleration rates 4 and 6. **Best values** are highlighted per metric. MSM achieves the best performance across both distortion-based and perception-oriented metrics.

| Setup | | Input | A-DPS | MSM ($w=3$ without noise reinject) | MSM ($w=3$ with noise reinject) |
|---|---|---|---|---|---|
| **Inpainting** | PSNR↑ | 18.26 | 20.14 | 24.63 | **24.71** |
| | SSIM↑ | 0.749 | 0.621 | 0.867 | **0.867** |
| | LPIPS↓ | 0.304 | 0.305 | 0.076 | **0.076** |
| **SR** ($\times 4$) | PSNR↑ | 23.21 | 22.61 | 27.99 | **28.11** |
| | SSIM↑ | 0.728 | 0.702 | 0.866 | **0.868** |
| | LPIPS↓ | 0.459 | 0.277 | 0.117 | **0.117** |
| **CS-MRI** ($\times 4$) | PSNR↑ | 22.75 | 27.28 | 29.71 | **30.71** |
| | SSIM↑ | 0.648 | 0.804 | 0.814 | **0.839** |
| | LPIPS↓ | 0.306 | 0.173 | 0.164 | **0.145** |
| **CS-MRI** ($\times 6$) | PSNR↑ | 21.94 | 26.29 | 27.67 | **28.86** |
| | SSIM↑ | 0.617 | 0.763 | 0.769 | **0.805** |
| | LPIPS↓ | 0.342 | 0.201 | 0.193 | **0.168** |

### E.9 Learning Partial Measurement Scores from Noisy Measurements with Incorrect Noise Assumptions

In practice, the noise level of the measurements is rarely known exactly, even though prior work on training diffusion models from noisy or incomplete data typically assumes access to a noise estimate (Aali et al., 2023; Chen et al., 2022; Daras et al., 2024b; Kawar et al., 2024). To assess MSM's sensitivity to this assumption, we evaluate the noisy-measurement training procedure from Section 3.4 under both a matched and mismatched setting. The model is trained assuming a measurement noise level of $\rho_{\text{assumed}} = 0.1$ and we compare its performance when the true test-time noise matches this value ($\rho_{\text{true}} = 0.1$) versus when it is lower ($\rho_{\text{true}} = 0.05$). As shown in Table 16, MSM exhibits only a minor change in FID, indicating that it is robust to moderate mis-specification of the measurement noise level during training.

Table 16: FID scores for unconditional generation under noise-level mis-specification. The model is trained assuming $\rho_{\text{assumed}} = 0.1$, while the true noise level $\rho_{\text{true}}$ may differ. MSM shows only minor degradation when the assumed noise level does not match the actual one, demonstrating robustness to noise level mis-specification.

| Training data | Methods | FID↓ |
|---|---|---|
| $p = 0.4, \rho_{\text{assumed}} = 0.1, \rho_{\text{true}} = 0.1$ | MSM (Matched) | 37.14 |
| $p = 0.4, \rho_{\text{assumed}} = 0.1, \rho_{\text{true}} = 0.05$ | MSM (Mismatched) | 38.28 |

### E.10 COMPARISON OF INFERENCE EFFICIENCY FOR CONDITIONAL SAMPLING: MSM VS. AMBIENT DPS

We compare the inference efficiency of MSM and Ambient DPS for conditional sampling in inverse problems. Since MSM requires $w$ stochastic loops per diffusion step while Ambient DPS does not, it is important to assess this cost–performance trade-off. All methods are evaluated under the 200 diffusion sampling steps. We test MSM with $w \in \{1, 2, 3\}$ across inpainting, super-resolution, and compressed-sensing MRI. Even with the minimal configuration $w = 1$, MSM achieves higher reconstruction quality than Ambient DPS while requiring only a single loop per diffusion step. Increasing $w$ provides the expected accuracy–efficiency trade-off characteristic of our stochastic score aggregation. Results are summarized in Table 17.

Table 17: Inference efficiency comparison between Ambient DPS and MSM under different stochastic loop $w$.

| Setup | | Input | A-DPS | MSM ($w = 1$) | MSM ($w = 2$) | MSM ($w = 3$) |
|---|---|---|---|---|---|---|
| **Inpainting** | PSNR↑ | 18.26 | 20.14 | 24.48 | 24.64 | 24.70 |
| | SSIM↑ | 0.749 | 0.621 | 0.868 | 0.867 | 0.868 |
| | LPIPS↓ | 0.304 | 0.305 | 0.075 | 0.076 | 0.075 |
| **SR** ($\times 4$) | PSNR↑ | 23.21 | 22.61 | 28.27 | 28.15 | 28.09 |
| | SSIM↑ | 0.728 | 0.702 | 0.872 | 0.870 | 0.869 |
| | LPIPS↓ | 0.459 | 0.277 | 0.120 | 0.117 | 0.116 |
| **CS-MRI** ($\times 4$) | PSNR↑ | 22.75 | 27.28 | 29.12 | 30.50 | 30.71 |
| | SSIM↑ | 0.648 | 0.804 | 0.806 | 0.828 | 0.839 |
| | LPIPS↓ | 0.306 | 0.173 | 0.172 | 0.154 | 0.145 |
| **CS-MRI** ($\times 6$) | PSNR↑ | 21.94 | 26.29 | 27.20 | 28.21 | 28.86 |
| | SSIM↑ | 0.617 | 0.763 | 0.759 | 0.789 | 0.805 |
| | LPIPS↓ | 0.342 | 0.201 | 0.202 | 0.178 | 0.168 |

## F    USE OF LARGE LANGUAGE MODELS

In preparing this manuscript, we used large language models (LLMs) solely for minor editorial assistance, such as correcting grammar and fixing typographical errors. LLMs were not used for research ideation, methodological development, data analysis, or substantive writing. All scientific contributions and writing decisions are entirely those of the authors.

## G    REPRODUCIBILITY STATEMENT

We have made significant efforts to ensure reproducibility of our work. An anonymous supplementary code package is provided, which fully reproduces our proposed method for both training and unconditional sampling, as well as for solving inverse problems. Detailed descriptions of the models, training configurations, and evaluation protocols are included in the main paper and Appendix D.4. Additional implementation details and data preprocessing steps are provided in the supplementary materials.

