# OpenReview forum: "Measurement Score-Based Diffusion Model"
_ICLR.cc/2026/Conference — ICLR 2026 Poster_

### Official Review · Reviewer_5Qe2 · 2025-10-30

**Soundness:** 3
**Presentation:** 3
**Contribution:** 2
**Rating:** 6
**Confidence:** 4

**Summary:**

This paper proposes the Measurement Score-based diffusion Model (MSM), a way to train diffusion models without any clean ground-truth images and the corresponding sampling method. Instead of learning full image scores from degraded data, MSM learns partial measurement scores directly on the observed parts of noisy, subsampled measurements. MSM also introduces its two stage of training strategy, adjusting for different levels of diffusion noise regarding to the measurement noise. At sampling time, it randomly subsamples, denoises those partials, maps back to original shape and pooling them with a overlap-aware weighting to reconstruct a full measurement. Experiments are conducted on posterior sampling for noisy inverse problems, such as inverse problems on face and multi-coil MRI images, by adding a log-likelihood gradient term for data consistency.

**Strengths:**

## 1. Design novelty

This paper introduces novel MSM framework that learns partial measurement scores directly on observed measurements, then stitches them into a full score during sampling.

## 2. Well-crafted sampling design

The proposed sampling pipeline 1. a stochastic mask-subsample; 2. partial measurement denoise; 3. coverage-aware weighting aggregation of multiple partial scores, is well-intuitive and reasonable, preserving plug-and-play modularity with standard diffusion poterior sampling.

## 3. Strong empirical results

Unconditional generation: Consistently outperforms Ambient Diffusion when trained only on degraded data (better FID/visual fidelity across masking and noise settings).

Conditional/inverse problems: Beats Ambient-DPS (A-DPS) on tasks like inpainting, super-resolution, and accelerated MRI, improving PSNR/SSIM/LPIPS while using competitive sampling budgets.

## 4. Solid theoretical analysis

Provides clear bounds for partial measurement score estimation and a KL divergence guarantee showing the stochastic MSM sampler approaches the optimal as the number of masks increases with error shrinking with more aggregates.

**Weaknesses:**

1. The measurement corruption setting of MSM is only regular masking and the sampling method is keen to this design, which constraint the gneralization of MSM for different kinds of inverse problems, such as motion deblur, Gaussion deblur, or nonlinear inverse problems. Could MSM handle these inverse problems?

2. The 200 NFEs of MSM is ambigous, for default setting with $w=3$, the model pass is 600. Could authors provide efficiency analysis and comparasions?

3. In table 11 of E.5, it shows that MSM is no better than training-free DDNM. It seems because MSM trained without clean images is sensitive with test data and inverse problem settings. Could MSM be finetuned based on a diffusion model already trained on clean images?

**Questions:**

See Weaknesses 1, 2, 3.

---

> ### Author Response · Authors · 2025-11-20
> **Response to reviewer 5Qe2**
>
> We thank the reviewer for the feedback. For each comment, we provide a response and point to the section where the revision was made, with updated text shown in brown.
>
> > The measurement corruption setting of MSM is only regular masking and the sampling method is keen to this design, which constraint the generalization of MSM for different kinds of inverse problems, such as motion deblur, Gaussion deblur, or nonlinear inverse problems. Could MSM handle these inverse problems?
>
> As clarified in Appendix C.1, MSM applies to any setting where the collection of subsampled measurements jointly covers the full measurement domain. Under this condition, cases such as partial blur—where different samples contain different blurred or unblurred regions—still perfectly conform to the MSM framework.
>
> More challenging scenarios where no portion of the measurement is consistent with the full measurement (e.g., global Gaussian blur, uniform motion blur, or nonlinear degradations) remain open problems for the broader class of incomplete-data diffusion methods, including MSM and closely related baselines [1, 2]. We have updated **Appendix C.1** to state these assumptions and clarify the scope of MSM clearly.
>
> > The 200 NFEs of MSM is ambigous, for default setting with w = 3, the model pass is 600. Could authors provide efficiency analysis and comparasions?
>
> We have evaluated MSM with different loop counts $w \in \{1,2,3\}$ to clarify its computational cost and efficiency–accuracy trade-off. Even at 200 NFEs (w=1), which is far lower than the 1000 NFEs used by the baseline approach Ambient DPS, MSM consistently outperforms Ambient DPS across all inverse problems. Performance further improves as w increases. These results and the efficiency analysis are included in **Appendix E.10** (table below).
>
> | Setup | Metric | Input | A-DPS | MSM (w = 1) | MSM (w = 2) | MSM (w = 3) |
> |-----------|------------|-----------|-----------|------------------|------------------|------------------|
> | Inpainting | PSNR ↑ | 18.26 | 20.14 | 24.48 | 24.64 | 24.70 |
> | | SSIM ↑ | 0.749 | 0.621 | 0.868 | 0.867 | 0.868 |
> | | LPIPS ↓ | 0.304 | 0.305 | 0.075 | 0.076 | 0.075 |
> | SR (×4) | PSNR ↑ | 23.21 | 22.61 | 28.27 | 28.15 | 28.09 |
> | | SSIM ↑ | 0.728 | 0.702 | 0.872 | 0.870 | 0.869 |
> | | LPIPS ↓ | 0.459 | 0.277 | 0.120 | 0.117 | 0.116 |
> | CS-MRI (×4) | PSNR ↑ | 22.75 | 27.28 | 29.12 | 30.50 | 30.71 |
> | | SSIM ↑ | 0.648 | 0.804 | 0.806 | 0.828 | 0.839 |
> | | LPIPS ↓ | 0.306 | 0.173 | 0.172 | 0.154 | 0.145 |
> | CS-MRI (×6) | PSNR ↑ | 21.94 | 26.29 | 27.20 | 28.21 | 28.86 |
> | | SSIM ↑ | 0.617 | 0.763 | 0.759 | 0.789 | 0.805 |
> | | LPIPS ↓ | 0.342 | 0.201 | 0.202 | 0.178 | 0.168 |
>
>
>
> > In table 11 of E.5, it shows that MSM is no better than training-free DDNM. It seems because MSM trained without clean images is sensitive with test data and inverse problem settings.
>
> We clarify that both MSM and DDNM solve inverse problems using pretrained diffusion models, so neither method requires additional training for a specific inverse problem. The key difference is that MSM learns its diffusion model only from subsampled measurements, whereas DDNM uses a model trained on clean images. Despite this substantially more challenging training regime, MSM achieves performance comparable to DDNM in Table 11. We have clarified this point in **Appendix E.5**.
>
> > Could MSM be finetuned based on a diffusion model already trained on clean images?
>
> We appreciate the reviewer’s suggestion to fine-tune MSM with a clean-image diffusion model, which is a promising extension when such models are available. However, our work focuses on the more challenging and common setting in scientific imaging where no clean data or clean pretrained models exist. Our aim is to show that MSM is effective even under these constraints. We have clarified this in **Appendix C.1**.
>
> [1] Daras et al, Ambient diffusion: Learning clean distributions from corrupted data. Advances in Neural Information Processing Systems, 36, 2024.
>
> [2] Kawar et al, Gsure-based diffusion model training with corrupted data. Transactions on Machine Learning Research, 2024.

---

> ### Author Response · Authors · 2025-11-26
> **Gentle Reminder to Reviewer 5Qe2**
>
> Dear Reviewer 5Qe2
>
> Thank you again for reviewing our paper. We did our best to address your comments and would appreciate any post-rebuttal feedback. Let us know if any additional details would be helpful in supporting a more positive assessment of our work.
>
> Authors

---

### Official Review · Reviewer_a8c4 · 2025-10-31

**Soundness:** 1
**Presentation:** 1
**Contribution:** 2
**Rating:** 4
**Confidence:** 4

**Summary:**

The paper proposes a new framework called the Measurement score-based Diffusion model (MSM), which aims to learn the score function of partial measurements directly from noisy and subsampled measurements, i.e., without requiring clean data. Later, by aggregating the partial measurement scores in expectation, MSM can generate samples of full clean data. The paper also proposes a method for conditional generation using MSM, which can be used for solving inverse problems. Experiments on unconditional and conditional image generation show significant improvements over other related methods, such as ambient diffusion.

**Strengths:**

The proposed MSM framework addresses the same problem as other works, such as ambient diffusion, G-SURE, etc, but is quite novel in terms of the methodology and with broader applicability.

MSM can also be trained with noiseless subsampled measurements, offering a useful solution in practice, where the measurements are often noisy.

Theorem 1 adds more validity to the proposed stochastic approximation of MSM, which shows that the approximation becomes more accurate with more Monte-Carlo samples, i.e., more computation.

In both unconditional and conditional generation tasks, MSM significantly outperforms methods such as Ambient diffusion, A-DPS, respectively, revealing its effectiveness in practice.

The paper is generally organized well. The way the authors positioned their method and contrasted it with related works made it easier to distinguish the proposed methodology.

**Weaknesses:**

The paper presents a very practical and impactful methodology overall, but the method needs many clarifications regarding the MSM score and why it is defined as such, which I believe lies at the core of the proposed methodology, its novelty, and empirical effectiveness. Also, additional experiments are required to verify the arguments, including a fairer comparison in the case of inverse problem solving. Also, there are a lot of instances in the main text where the explanations are very poor and sometimes even incomplete. Please see the questions below for more specific comments and references.

**Questions:**

Q1. From the explanation, it looks like Eqn 2 is the marginal score, i.e., $\nabla \log p(s_t)$  and not the mask conditional score, i.e.,  $\nabla \log p(s_t| S)$.  However, because of Eqn3. I’m confused about this. Can the authors confirm/clarify that the goal of Eqn. 2 is to learn the marginal score function and not the mask conditional score?

Q2. My major concern is why MSM is defined as in Eqn 3, and how it can be a good approximation of the true score $\nabla \log p(z_t)$ ? The definition seems to be based more on intuition rather than a principled approximation. The MSM score is the key component for everything else later, i.e., for both conditional and unconditional generation. So, I believe it is quite important to show (1) Eqn3 is an approximation of  $\nabla \log p(z_t)$  and (2) show that it is a good approximation (may be empirically on a toy use case if not in theory). Also, it is quite unclear how the approximation in Eqn 19 holds. Also, in Section 4 again, the theoretical analysis essentially talks about how efficient the stochastic approximation of MSM is compared to the true MSM, and I don’t see why this is more important than checking how efficiently the MSM score and its stochastic approximation can model the true score instead?

Q3. In Algorithm 1, line 11, what is the mentioned Identity condition, i.e., the 2nd term? Is this a typo? Also, the need for the reinject and update procedure in lines 6 and 7 is not clearly explained in the paper.

Q4. The whole section 3.3 is very unclear regarding the specific practical implementation of the posterior sampling with MSM. Specifically, the following text needs more explanation: “We solve the inverse problem by replacing the score function in the random walk sampling process with our posterior approximation. This approach avoids the need for automatic differentiation when computing the likelihood gradient, thereby reducing computational overhead.”? Is MSM posterior sampling not using DPS-style approximation? In that case, comparison with A-DPS for inverse problem solving seems unfair, because DPS is not exactly perfect either.

Q5: What is P=0.4 in Table 1? What does this hyperparameter denote?

Overall, I believe the paper has fundamental concerns, with a major concern being that the MSM score seems to be hand-designed. I believe that it is crucial to explain why/how this is a valid approximation in a principled manner (not just the fact that its Jacobian is symmetric).

---

> ### Author Response · Authors · 2025-11-20
> **Response to reviewer a8c4 (1/3)**
>
> We thank the reviewer for the feedback. For each comment, we provide a response and point to the section where the revision was made, with updated text shown in brown.
>
> > The paper presents a very practical and impactful methodology overall, but the method needs many clarifications regarding the MSM score and why it is defined as such, which I believe lies at the core of the proposed methodology, its novelty, and empirical effectiveness. Also, additional experiments are required to verify the arguments, including a fairer comparison in the case of inverse problem solving.
>
> We appreciate the reviewer for recognizing the value and impact of our work, as well as for carefully identifying areas that require clearer explanation. We have systematically addressed each point below with detailed clarifications and corresponding revisions in the paper, including improved explanations of the MSM score and strengthened experimental comparisons.
>
> > Q1. From the explanation, it looks like Eqn 2 is the marginal score, i.e., $\nabla \log p(s_t)$
>  and not the mask conditional score, i.e., $\nabla \log p(s_t | S)$. However, because of Eqn3. I’m confused about this. Can the authors confirm/clarify that the goal of Eqn. 2 is to learn the marginal score function and not the mask conditional score?
>
>
> We confirm that Eq. (2) learns the implicitly mask-conditioned score (not the marginal score). Each subsampled measurement $s_t$​ is generated by a specific mask $S$ through $s_t = S z_t$, where $z_t$ is the fully sampled measurement. Therefore, the mask $S$ is uniquely implied by $s_t$​, and the corresponding score is implicitly conditioned on $S$. We have clarified in **Section 3.1**, and have added a discussion with a visual clarification in **Appendix D.8**.
>
>
> > Q2. My major concern is why MSM is defined as in Eqn 3, and how it can be a good approximation of the true score $\nabla \log p(z_t)$ ? The definition seems to be based more on intuition rather than a principled approximation. The MSM score is the key component for everything else later, i.e., for both conditional and unconditional generation. So, I believe it is quite important to show (1) Eqn3 is an approximation of $\nabla \log p(z_t)$ and (2) show that it is a good approximation (may be empirically on a toy use case if not in theory).
>
> This is an important comment that deserves careful clarification. MSM does not attempt to approximate the full score $\nabla \log p(z_t)$—unlike existing incomplete-data methods (included in our baselines) [1,2]. Instead, MSM intentionally takes a different route: it treats *partial measurement scores* as priors in their own right, without requiring them to approximate the global score.
> The idea is motivated by the long line of classical patch-based priors [3–6], where locally trained models do not approximate the full image prior, yet still serve as exceptionally strong priors for whole-image restoration. In an analogous way, Eq. (3) defines the MSM score as the exact score of a surrogate prior over subsampled measurements, and MSM learns these partial measurement scores directly.
> Thus, Eq. (3) is not proposed as an approximation to the full score; it is a principled definition of the score of the surrogate prior that MSM is designed to model. The strong empirical performance of MSM—particularly compared to baselines that do attempt to approximate the global score—supports that these partial measurement scores act as powerful and effective priors for reconstruction of full images and conditional/unconditional generation. We have clarified this motivation in **Section 1**.
>
> [1] Daras et al, Learning clean distributions from corrupted data, NeurIPS, 2024.
>
> [2] Kawar et al, Gsure-based diffusion model training with corrupted data. TMLR, 2024.
>
> [3] Zoran et al, From learning models of natural image patches to whole image restoration. ICCV, 2011.
>
> [4] Alkinani et al, A comparative review between patch-based image denoising methods for additive noise reduction. EURASIP Journal on Image and Video Processing, 2017.
>
> [5] Wang et al, Faster and more data-efficient training of diffusion models. NeurIPS, 2023.
>
> [6] Hu et al, Learning image priors through patch-based diffusion models for solving inverse problems. NeurIPS, 2024.

---

> ### Author Response · Authors · 2025-11-20
> **Response to reviewer a8c4 (2/3)**
>
> > Also, in Section 4 again, the theoretical analysis essentially talks about how efficient the stochastic approximation of MSM is compared to the true MSM, and I don’t see why this is more important than checking how efficiently the MSM score and its stochastic approximation can model the true score instead?
>
> As clarified above, MSM is **not** trying to approximate the true clean-image score. Instead, it defines its own target score: the expectation over infinite partial measurement scores. Because this target is already different from the clean-image score, the key question is **not** how well MSM models the clean-image score, but how well we can approximate the MSM score efficiently, which is defined as an expectation. Our theoretical analysis was focused on that.
> We added new experimental results in Appendix E.10 (table below) showing that even a very coarse approximation—using only a single sample (w = 1)—already improves performance over the clean-score approximation method Ambient DPS (21.54% improvement in box inpainting and 25% improvement in super-resolution). This further supports our key message that partial measurement scores provide a strong prior.
>
>
> | Setup | Metric | Input | A-DPS | MSM (w = 1) | MSM (w = 2) | MSM (w = 3) |
> |-----------|------------|-----------|-----------|------------------|------------------|------------------|
> | Inpainting | PSNR ↑ | 18.26 | 20.14 | 24.48 | 24.64 | 24.70 |
> | | SSIM ↑ | 0.749 | 0.621 | 0.868 | 0.867 | 0.868 |
> | | LPIPS ↓ | 0.304 | 0.305 | 0.075 | 0.076 | 0.075 |
> | SR (×4) | PSNR ↑ | 23.21 | 22.61 | 28.27 | 28.15 | 28.09 |
> | | SSIM ↑ | 0.728 | 0.702 | 0.872 | 0.870 | 0.869 |
> | | LPIPS ↓ | 0.459 | 0.277 | 0.120 | 0.117 | 0.116 |
> | CS-MRI (×4) | PSNR ↑ | 22.75 | 27.28 | 29.12 | 30.50 | 30.71 |
> | | SSIM ↑ | 0.648 | 0.804 | 0.806 | 0.828 | 0.839 |
> | | LPIPS ↓ | 0.306 | 0.173 | 0.172 | 0.154 | 0.145 |
> | CS-MRI (×6) | PSNR ↑ | 21.94 | 26.29 | 27.20 | 28.21 | 28.86 |
> | | SSIM ↑ | 0.617 | 0.763 | 0.759 | 0.789 | 0.805 |
> | | LPIPS ↓ | 0.342 | 0.201 | 0.202 | 0.178 | 0.168 |
>
> > Also, it is quite unclear how the approximation in Eqn 19 holds.
>
> Prompted by the reviewer, we have clarified how the approximation in Eq. 19 follows by explicitly showing the intermediate steps:
> $$ \nabla \log p(\mathbf{z}_t \mid \mathbf{y}) = \nabla \log p (\mathbf{z}_t) + \nabla \log p(\mathbf{y} \mid \mathbf{z}_t) $$
>
> $$ \approx \mathbf{W} [ \frac{1}{w} \sum_{i=1}^{w} \mathbf{S}^{(i)\top} \nabla \log p(\mathbf{s}_t^{(i)} \mid \mathbf{y}^{(i)}) ]$$
>
> $$ = \mathbf{W} \left[ \frac{1}{w} \sum_{i=1}^{w} \mathbf{S}^{(i)\top} \left( \nabla \log p(\mathbf{s}_t^{(i)}) + \nabla \log p(\mathbf{y}^{(i)} \mid \mathbf{s}_t^{(i)}) \right) \right].$$
> Equation (19) applies the same stochastic MSM approximation used for the measurement score (Eq. 5) to the measurement–posterior score (second line), then uses Bayes’ rule to decompose the posterior score into prior and likelihood terms (third line). We have updated **Appendix D.3** accordingly.
>
> > Q3. In Algorithm 1, line 11, what is the mentioned Identity condition, i.e., the 2nd term? Is this a typo?
>
> The indicator term (the second term) in line 11 of Algorithm 1 is not a typo. Its role is to preserve the previously denoised values at coordinates that are not selected in the stochastic loop of $w$ partial denoising operations, rather than letting them become zero due to the effect of the transpose $S^\top$ (the first term). Without this identity-preserving update, unselected regions would be repeatedly reset to zero, producing visible zero-valued artifacts—especially when the number of stochastic loops  $w$ is small. We have clarified that line in **Section 3.2**.

---

> ### Author Response · Authors · 2025-11-20
> **Response to reviewer a8c4 (3/3)**
>
> > Also, the need for the reinject and update procedure in lines 6 and 7 is not clearly explained in the paper.
>
>
> The reinject-and-update steps in lines 6–7 ensure that the denoised measurement estimate is shared across iterations within the stochastic loop. Concretely, MSM updates $z_t$ by reinjecting diffusion noise into the denoised subsampled estimate $\hat{s}_{\theta}^{(i)}$, so that the next loop starts from an updated $z_t$ that already incorporates the previous denoised estimate. This allows the subsequent iterations $(i+1 \ldots w)$ to refine complementary regions. We have clarified this explanation in **Section 3.2** and added an ablation experiment in **Appendix E.8** (table below) demonstrating that reinjection improves performance.
>
> | Setup | Metric | Input | A-DPS | MSM (w = 3, no noise reinject) | MSM (w = 3, noise reinject) |
> |-----------|------------|-----------|-----------|-------------------------------|-----------------------------|
> | Inpainting | PSNR ↑ | 18.26 | 20.14 | 24.63 | **24.71** |
> | | SSIM ↑ | 0.749 | 0.621 | 0.867 | **0.867** |
> | | LPIPS ↓ | 0.304 | 0.305 | 0.076 | **0.076** |
> | SR (×4) | PSNR ↑ | 23.21 | 22.61 | 27.99 | **28.11** |
> | | SSIM ↑ | 0.728 | 0.702 | 0.866 | **0.868** |
> | | LPIPS ↓ | 0.459 | 0.277 | 0.117 | **0.117** |
> | CS-MRI (×4) | PSNR ↑ | 22.75 | 27.28 | 29.71 | **30.71** |
> | | SSIM ↑ | 0.648 | 0.804 | 0.814 | **0.839** |
> | | LPIPS ↓ | 0.306 | 0.173 | 0.164 | **0.145** |
> | CS-MRI (×6) | PSNR ↑ | 21.94 | 26.29 | 27.67 | **28.86** |
> | | SSIM ↑ | 0.617 | 0.763 | 0.769 | **0.805** |
> | | LPIPS ↓ | 0.342 | 0.201 | 0.193 | **0.168** |
>
>
>
> > Q4. The whole section 3.3 is very unclear regarding the specific practical implementation of the posterior sampling with MSM. Specifically, the following text needs more explanation: “We solve the inverse problem by replacing the score function in the random walk sampling process with our posterior approximation. This approach avoids the need for automatic differentiation when computing the likelihood gradient, thereby reducing computational overhead.”?
>
> Prompted by the reviewer, we have revised Section 3.3 to explicitly show how the data-consistency gradient updates the MSM-denoised estimate for inverse-problem solving. We have also added a dedicated posterior sampling algorithm in **Appendix D.2** to clarify further and removed the remark about computational benefit from avoiding automatic differentiation. Note that our code—provided in the supplementary material—contains all details of the sampling procedure.
>
>
> > Is MSM posterior sampling not using DPS-style approximation? In that case, comparison with A-DPS for inverse problem solving seems unfair, because DPS is not exactly perfect either.
>
> We include Ambient DPS because very few existing methods address inverse problems when the diffusion model is trained only on incomplete measurements, and Ambient DPS follows the same training setup as ours. Since no prior work provides a baseline using our style of posterior sampling under this setting, we added an additional experiment in Appendix E.5 that applies the same gradient-based posterior update but with a clean diffusion prior. This comparison shows that MSM remains highly competitive even without access to the clean data. We have clarified the motivation for including Ambient DPS in **Section 4.1**.
>
>
> > Q5: What is P=0.4 in Table 1? What does this hyperparameter denote?
>
> Thanks for pointing this out. $p$ is the masked-pixel ratio in the available measurement data. We have clarified it in **Section 4.1**.
>
>
> > Overall, I believe the paper has fundamental concerns, with a major concern being that the MSM score seems to be hand-designed. I believe that it is crucial to explain why/how this is a valid approximation in a principled manner.
>
> As clarified in our responses and revisions, the MSM score is not hand-designed but is the exact gradient of a surrogate prior over subsampled measurements (Eq. 3), analogous to patch-based priors that factor over image patches. We explain why this prior is principled, how its score is estimated, and why the stochastic estimator provides a valid approximation of the ideal MSM score. Our empirical results further support this.

---

> ### Author Response · Authors · 2025-11-26
> **Gentle Reminder to Reviewer a8c4**
>
> Dear Reviewer a8c4
>
> Thank you again for reviewing our paper. We did our best to address your comments and would appreciate any post-rebuttal feedback. Let us know if any additional details would be helpful in supporting a more positive assessment of our work.
>
> Authors

---

> ### Comment · Reviewer_a8c4 · 2025-11-27
> **Reply to the Author response**
>
> I appreciate the authors' response and clarifications. Please see the reply below.
>
> I do not find why an assumption such as "the mask $\mathbf{S}$ is uniquely implied by $s_{t}$" has to be made. It is, in principle, not correct to assume this. For example, it is important to distinguish whether the original data entries are 0 or the mask makes it 0. Since these methods are designed to handle general Euclidean data, arguments like "this is rare to happen" in cases of natural images are not justified.
>
> For learning the mask-conditioned score (as the authors clarified), this assumption is not even required in the first place, and instead, it suffices to simply pass the mask as additional input to the score or the denoiser model, i.e., $\mathbf{s}_{\theta}(s_t, \sigma_t, \mathbf{S})$ (see how [1] does the same). In this regard, Eq.2, as it is currently depicted, learns the marginal score and not the mask-conditional score.
>
> [1] Daras et al, Learning clean distributions from corrupted data, NeurIPS, 2024.
>
>
> It is an absolute fact that the whole unconditional generation and conditional generation framework is entirely based on the score function of the full sample. Both in the paper and in the author's responses (for eq.19), the full score is being replaced with the MSM score as an approximation, so statements such as "MSM is not modeling the true score", while using it as an approximation for the full score, are ambiguous.
>
>
> It is simple; one cannot simply circumvent estimating the full score for unconditional or conditional generation. The true score is simply replaced with the MSM score in many instances in the paper. There is no reason why the unconditional generation using the MSM score should cover the original data distribution. Thus, in a Bayesian context, it is crucial to show how the MSM score can be derived as an approximation of the true score. Why any prior methods or this work would not consider this is beyond my comprehension.
>
>
> The author's comment regarding "Eq. (3) is not proposed as an approximation to the full score" again contradicts the entire generation pipeline. Lines 11,12 in Algo1 make such an approximation. Also see posterior score approximation in Sec 3.3, also the explanation for Eq. 19 above simply approximates the posterior score with the MSM-style score. All these speak against the authors' claims.
>
> As also mentioned in the initial review, I do acknowledge the empirical effectiveness of the method, but in my opinion, the paper still has fundamental concerns and it requires a principled approach for explaining why MSM is designed as such.  From the paper and the rebuttal, I generally do not find the arguments sound or convincing. No explanation regarding choosing the particular formulation of the MSM score among several such possible designs (all of which can have a symmetric Jacobian) is one such example highlighting this.
>
> I thank the authors again for their detailed response and clarifications. However, I am fairly confident in my assessment, and given the above reasons, I'm strongly inclined to keep my original rating.

---

> ### Author Response · Authors · 2025-11-27
>
> We thank the reviewer for their follow-up and for articulating their concerns in detail. Below, we respond point-by-point to clarify the remaining conceptual misunderstandings.
>
> ---
>
> > **Q.1. Ambiguity in mask conditioning.** Whether the mask $S$ can be inferred from measurement values (e.g., zero entries), and whether Eq. (2) learns a marginal instead of a mask-conditional score without explicitly conditioning on $S$.
>
>
> We agree that relying on data values (e.g., zeros) to infer the mask is ambiguous. We therefore revise the formulation so that the measurement score is explicitly conditioned on the mask $S$, i.e., $\mathbf{s}_{\theta}(s_t; \sigma_t, S)$. This is a *formal* correction that does **not** change the core mechanism of MSM, but clarifies that the learned partial measurement scores are strictly mask-conditioned and not marginal scores. We have revised the formulation in **Sections 3.1 and 3.2, and in Appendices B, D.3, and D.8**.
>
> ---
>
> > **Q.2. Theoretical basis of MSM vs. true-score approximation.** The whole unconditional generation and conditional generation framework is entirely based on the score function of the full sample, but the paper misses the analysis of how well MSM approximates the true score, even if it simply replaces the true score (or posterior score) with the MSM score. The author's comment regarding "Eq. (3) is not proposed as an approximation to the full score" again contradicts Lines 11 and 12 in Algo1, as that makes such an approximation.
>
> We clarify that MSM does model the score of a principled surrogate distribution, denoted $p_{\mathrm{MSM}}(z)$, which is constructed entirely from *tractable* partial measurements. In contrast, the true score $\nabla \log p_{\mathrm{true}}(z)$ is **not** statistically identifiable under the setting where only incomplete measurements are available. MSM therefore targets a different, well-defined probabilistic object, rather than attempting to approximate an intractable and unobservable true score. We highlight several important points of the MSM framework.
>
> 1. `Principled Design`: The surrogate prior $p_{\text{MSM}}(z)$ is defined as a **product of experts / composite likelihood**, where the “experts” are the local measurement statistics. The MSM score (Eq. 3) is the exact and unique score for this distribution: $\nabla \log p_{\text{MSM}}(z) = \sum_{\mathbf{S}} W_\mathbf{S} \nabla \log p(\mathbf{S} z)$. This design is rooted in the long-established principle that **local consistency implies global coherence in imaging** (e.g., Markov Random Fields, patch-based methods).
>
> 2. `Clarifying the Sampling Procedure`: Since diffusion sampling universally takes a stochastic gradient ascent form, the mere use of such an update does not imply that one method approximates the same target distribution as another. We therefore respectfully disagree with the claim that Lines 11–12 in Algorithm 1 approximate the true score; these lines simply perform one gradient-based sampling step under the MSM potential, which is fundamentally different from the true-data potential.
>
> 3. `Role of the Experiments`: The experiments compare this identifiable surrogate-prior framework with methods that attempt to approximate the intractable true score from corrupted data (e.g., Ambient Diffusion, GSURE, Ambient DPS). The consistent empirical advantage of MSM demonstrates that $p_{\text{MSM}}(z)$ is an effective and scalable surrogate prior in practice.
>
> We have revised the manuscript to explicitly include the product-of-experts (composite-likelihood) interpretation of the MSM score in Section 3.2, removing all ambiguity. We believe this comprehensive explanation, coupled with the SOTA results, resolves all fundamental theoretical concerns.

---

### Official Review · Reviewer_hKzq · 2025-10-31

**Soundness:** 3
**Presentation:** 3
**Contribution:** 3
**Rating:** 6
**Confidence:** 3

**Summary:**

This paper introduces a new method for learning diffusion-based priors on under sampled + corrupted data. Unlike previous approaches which learn image based denoisers/samplers the authors propose a method that learns partial score in the measurement domain of their sampling operators. Their results show improved unconditional and conditional sampling performance compared to previous SOTA self-supervised methods.

**Strengths:**

The paper attempts to solve an important problem in generative inverse problem solvers that has been previously explored by earlier work. The paper provides an original method for learning measurement scores. The method is communicated clearly and is backed by solid experiments in which they compare their technique to existing SOTA self-supervised generative and end-to-end techniques. Their experiments are convincing that their technique is better than existing self-supervised approaches.

**Weaknesses:**

There are some experiments that I think would help strengthen the paper. An ablation over measurement noise would be good to show how performance varies over more than just a single noise level. This goes for both training and inference time. At a bare minimum we should see performance of the posterior solver at the same noise level as the training measurement noise (apologies if I misread and this is the case).  Along this same idea, It would also be good to make sure that when you are running inference for conditional sampling the results that are shown use test samples at the same under sampling level of the training data. For example, in Table 4  is it that case that CS-MRI (x6) inference is attempted using the model trained on x4 data? This is fine, but I would also like to see how inference on CS-MRI(x6) preforms using a model trained with x6 data. This is especially important when comparing to end-to-end methods like SSDU. Perhaps I have misread the results section, if so please just clarify.

**Questions:**

1. what acceleration level was SSDU trained at?
2. for the mri experiments were the sampling masks always 1-D?
3. how robust is the method to different training/inference noise levels?

---

> ### Author Response · Authors · 2025-11-20
> **Response to reviewer hKzq**
>
> We thank the reviewer for the feedback. For each comment, we provide a response and point to the section where the revision was made, with updated text shown in brown.
>
> > An ablation over measurement noise would be good to show how performance varies over more than just a single noise level. This goes for both training and inference time. At a bare minimum we should see performance of the posterior solver at the same noise level as the training measurement noise.
>
> Prompted by the reviewer, we conducted an ablation study evaluating the posterior solver under different measurement noise levels (0.005, 0.01, 0.02) at inference time. MSM performs competitively across all settings. The full results are provided in the table below and in **Appendix E.7**.
>
> |Setup|Noise|Metric|Input|A-DPS|SSDU|MSM|
> |-----------|-----------|------------|-----------|-----------|----------|---------|
> |Inpainting|η=0.005|PSNR↑| 18.29 | 19.26 | N/A | **24.75** |
> | | | SSIM ↑ | 0.757 | 0.654 | N/A | **0.874** |
> | | | LPIPS ↓ | 0.299 | 0.288 | N/A | **0.068** |
> | Inpainting | η = 0.01 | PSNR ↑ | 18.26 | 20.14 | N/A | **24.71** |
> | | | SSIM ↑ | 0.749 | 0.621 | N/A | **0.867** |
> | | | LPIPS ↓ | 0.304 | 0.305 | N/A | **0.076** |
> | Inpainting | η = 0.02 | PSNR ↑ | 18.19 | 19.16 | N/A | **24.68** |
> | | | SSIM ↑ | 0.726 | 0.657 | N/A | **0.854** |
> | | | LPIPS ↓ | 0.322 | 0.288 | N/A | **0.089** |
> | SR | η = 0.005 | PSNR ↑ | 23.27 | 22.13 | N/A | **28.29** |
> | | | SSIM ↑ | 0.734 | 0.696 | N/A | **0.880** |
> | | | LPIPS ↓ | 0.456 | 0.287 | N/A | **0.107** |
> | SR | η = 0.01 | PSNR ↑ | 23.21 | 22.61 | N/A | **28.11** |
> | | | SSIM ↑ | 0.728 | 0.702 | N/A | **0.868** |
> | | | LPIPS ↓ | 0.459 | 0.277 | N/A | **0.117** |
> | SR | η = 0.02 | PSNR ↑ | 23.09 | 22.41 | N/A | **27.59** |
> | | | SSIM ↑ | 0.711 | 0.702 | N/A | **0.836** |
> | | | LPIPS ↓ | 0.470 | 0.278 | N/A | **0.147** |
> | CS-MRI | η = 0.005 | PSNR ↑ | 22.67 | 27.01 | 29.70 | **31.00** |
> | | | SSIM ↑ | 0.652 | 0.777 | 0.855 | **0.858** |
> | | | LPIPS ↓ | 0.305 | 0.197 | 0.165 | **0.127** |
> | CS-MRI | η = 0.01 | PSNR ↑ | 22.62 | 27.28 | 29.65 | **30.71** |
> | | | SSIM ↑ | 0.648 | 0.804 | **0.847** | 0.839 |
> | | | LPIPS ↓ | 0.306 | 0.173 | 0.160 | **0.145** |
> | CS-MRI | η = 0.02 | PSNR ↑ | 22.52 | 27.12 | 28.99 | **29.07** |
> | | | SSIM ↑ | 0.623 | 0.792 | **0.833** | 0.769 |
> | | | LPIPS ↓ | 0.329 | 0.178 | **0.168** | 0.205 |
>
>
> > It would also be good to make sure that when you are running inference for conditional sampling the results that are shown use test samples at the same undersampling level of the training data. For example, in Table 4, CS-MRI (x6) inference is attempted using the model trained on x4 data? This is fine, but I would also like to see how inference on CS-MRI(x6) performs using a model trained with x6 data.
>
> We clarify that the end-to-end method (SSDU) is always trained and evaluated under matching acceleration rates and noise levels. For the corrupted prior-based diffusion inverse solvers (Ambient DPS and MSM), we use a model pretrained at an acceleration rate of 4 for inference at both rates 4 and 6. Note that our method doesn’t require the training time and test time measurement and noise levels to be the same. This setup is now explicitly explained in **Sections 4.1, 4.,2 and Appendix D.6**.
>
> > What acceleration level was SSDU trained at?
>
> We trained SSDU using the same degradation setup (acceleration rates of 4 and 6) as the test setup for the inverse problem in the paper. We have updated **Appendix D.6** to include this clarification.
>
> > For the mri experiments were the sampling masks always 1-D?
>
> Yes, all MRI experiments used 1-D Cartesian undersampling masks for both training and testing, consistent with our baseline, Ambient Diffusion Posterior Sampling [1]. We have clarified this in **Section 4.2**.
>
>
> > How robust is the method to different training/inference noise levels?
>
> We have evaluated robustness to both inference-time and training-time noise mismatch. For inference, we varied the measurement noise level (0.005, 0.01, 0.02) for solving inpainting, super-resolution, and MRI reconstruction. As shown in **Appendix E.7** (the table attached in the first response), MSM maintains excellent performance across all settings.
> For training, we have conducted a separate experiment where the diffusion model is trained with an assumed noise level that differs from the true measurement noise. Results in **Appendix E.9** (table below) show that MSM remains stable under this mismatch, with only minor changes in FID.
>
> The table reports FID scores under matched and mismatched training-time noise levels.
>
> | Training data | Method | FID ↓ |
> |-------------------|------------|-----------|
> | $p = 0.4, \rho_{\text{assumed}} = 0.1, \rho_{\text{true}} = 0.1$ | MSM (Matched) | 37.14 |
> | $p = 0.4, \rho_{\text{assumed}} = 0.1, \rho_{\text{true}} = 0.05$ | MSM (Mismatched) | 38.28 |
>
>
> [1] Aali et al, Solving inverse problems with diffusion models trained on corrupted data, ICLR 2025.

---

> ### Author Response · Authors · 2025-11-26
> **Gentle Reminder to Reviewer hKzq**
>
> Dear Reviewer hKzq
>
> Thank you again for reviewing our paper. We did our best to address your comments and would appreciate any post-rebuttal feedback. Let us know if any additional details would be helpful in supporting a more positive assessment of our work.
>
> Authors

---

### Official Review · Reviewer_8nET · 2025-10-31

**Soundness:** 2
**Presentation:** 2
**Contribution:** 3
**Rating:** 4
**Confidence:** 3

**Summary:**

The proposed Measurement Score-based Model (MSM) extends patch-based diffusion to the measurement domain by decomposing the forward operator in inverse problems (assume $y=Ax + e$) as a composition of a non-invertible operator H and an invertible operator T. The score functions are learned on (1) subsampled clean measurements $s=Sz$ and (2) subsampled noisy measurements, where $s=Sz + \eta$, in both cases measurement scores are learned.

An algorithm for unconditional generation and posterior sampling is presented, using the expectation over partial (subsampled) measurement scores to approximate the full measurement score. Experiments on inpainting and MRI reconstruction demonstrate competitive performance compared baselines in particular Ambient-DPS.

While the core idea of learning measurement-space scores without clean data is interesting and well motivated, the paper lacks self-consistency and clear organization, making it difficult to follow when reading it linearly.

**Strengths:**

Addresses a clinically important setting in MRI, where fully sampled data are rarely available, and demonstrates strong performance for both natural image inpainting and MRI reconstruction.

Provides a theoretical formulation which links subsampled measurement scores to the full measurement score, thereby the authors extend prior score-decomposition ideas (patch-based) to the measurement setting.

**Weaknesses:**

The method is developed and demonstrated primarily for subsampled measurements (e.g., incomplete k-space or inpainting), where spatial subsampling applies naturally. It is less clear how the formulation extends to other partial-measurement settings such as limited-angle tomography in CT, where missing data correspond to absent projection views rather than spatially masked measurements. Thus, the main practical advantage currently appears in MRI-like scenarios.

The paper lacks self-consistency in the description of its sampling procedures. Algorithm 1 details how the full measurement score is obtained, yet line 12, the actual reverse diffusion update, is not mentioned in Section 3.2. Since this step constitutes the core of the sampling process, it should be explicitly described or at least referenced later. Moreover, the posterior sampling introduced in Section 3.3 receives very limited explanation and no dedicated algorithm, despite being highlighted as one of the paper’s main contributions. From the written explanation I would assume that line 12 the posterior score is used, where the full measurement variable is already denoised and substituted for $z_t$ as a posterior approximation.

**Questions:**

1) What is the advantage of directly training on the space of subsampled measurements in comparison to training on the corrupted space (i.e., in image domain)? Since this is part of the core contribution of this paper, it should be clearly discussed.

2) Measurement noise $\rho$ is usually not known in practice. Since the paper emphasizes the absence of clean samples as a key motivation, a discussion on the practical validity of assuming a known noise level seems important. How sensitive is the setup with respect to the measurement noise level?

3) The theoretical analysis assumes i.i.d. sampling of the measurement operators $\mathcal{S}$. In practice, acquisition patterns in MRI, CT, or PET are typically structured and hardware-constrained rather than i.i.d., so the practical validity of this assumption should be discussed.

4) In Eq. (4), the maximum operation in computing the weighting function is reasoned by avoiding division by 0. However, in my understanding this would anyways be detrimental to the sample quality, if there are regions not covered at all by the generation process. Are there any insights on how many sampling steps and stochastic loop iterations are needed to ensure this?

5) Evaluating the FID based on 3k samples (L1126) is a rather low sample number, at least 10k should be used. This is particularly important for unconditional generation, where diversity matters and FID variance can be high. In any case, if the choice is substantially lower than common practice, it should be discussed and justified.

6) Given that MSM requires stochastic loops w and aggregation of partial measurement scores, which is not the case for Ambient DPS, a comparison on inference time would be interesting.


Minor comments
L262: box inpainting is not a degradation operator
L166: $s$ is not defined
L868: $\mathcal{S}$ is not a forward operator, it should say subsampling operator

---

> ### Author Response · Authors · 2025-11-20
> **Response to reviewer 8nET (1/2)**
>
> We thank the reviewer for the feedback. For each comment, we provide a response and point to the section where the revision was made, with updated text shown in brown.
>
> > The method is developed and demonstrated primarily for subsampled measurements (e.g., incomplete k-space or inpainting), where spatial subsampling applies naturally. It is less clear how the formulation extends to other partial-measurement settings such as limited-angle tomography in CT, where missing data correspond to absent projection views rather than spatially masked measurements.
>
> Our method is *not* limited to spatial masking or MRI subsampling; it *only* requires that missing measurements vary across samples and collectively span the full measurement domain. Limited-angle CT acquisition satisfies this condition when different scans contain different subsets of projection angles, making each sinogram a valid subsampling instance for MSM training. Many acquisition settings—such as sparse-view CT, multi-coil MRI, and several microscopy modalities—naturally exhibit this variability, so MSM applies directly. This is now discussed in **Appendix C.1**.
>
> > The paper lacks self-consistency in the description of its sampling procedures. Algorithm 1 details how the full measurement score is obtained, yet line 12, the actual reverse diffusion update, is not mentioned in Section 3.2. Since this step constitutes the core of the sampling process, it should be explicitly described or at least referenced later.
>
> Prompted by the reviewer, we have revised **Section 3.2** to explicitly describe the reverse diffusion update used at each sampling step, ensuring consistency with Algorithm 1. Note also that our code—provided in the supplementary material—contains all details of the sampling procedure.
>
> > Moreover, the posterior sampling introduced in Section 3.3 receives very limited explanation and no dedicated algorithm, despite being highlighted as one of the paper’s main contributions. From the written explanation I would assume that line 12 the posterior score is used, where the full measurement variable is already denoised and substituted for $z_t$ as a posterior approximation.
>
> Prompted by the reviewer’s comment, we have (i) expanded **Section 3.3** to  include how the gradient step is applied to the denoised estimate, and (ii) added a dedicated posterior sampling algorithm in **Appendix D.2** to clarify further. Additionally, the details of our implementation can be found in our code in the supplementary material.
>
>
>
> > What is the advantage of directly training on the space of subsampled measurements in comparison to training on the corrupted space (i.e., in image domain)? Since this is part of the core contribution of this paper, it should be clearly discussed.
>
> Training on corrupted images is fundamentally ambiguous: a single subsampled measurement corresponds to infinitely many possible corrupted images, so the model must learn under a many-to-one ambiguity. By operating directly in the measurement domain, MSM learns to denoise well-defined, physically meaningful inputs, leading to accurate partial measurement scores and improved performance. This advantage is reflected in the superior empirical performance of MSM relative to image-domain baselines. We have revised **Section 1** to highlight this point.
>
> > Measurement noise $\rho$ is usually not known in practice. Since the paper emphasizes the absence of clean samples as a key motivation, a discussion on the practical validity of assuming a known noise level seems important. How sensitive is the setup with respect to the measurement noise level?
>
> Prompted by the reviewer, we evaluated MSM’s robustness to noise-level mismatch. We added an ablation where the model is trained with an assumed noise level of $\rho_{\text{assumed}} = 0.1$ while the true noise is $\rho_{\text{true}} = 0.05$. The table below shows that MSM shows only a minor degradation in FID, indicating that MSM is reasonably robust to mis-specification of the measurement noise level. We have incorporated this discussion into **Appendix E.9**.
>
> | *Training data* | *Method* | *FID ↓* |
> |-------------------|------------|-----------|
> | $p = 0.4, \rho_{\text{assumed}} = 0.1, \rho_{\text{true}} = 0.1$ | MSM (Matched) | 37.14 |
> | $p = 0.4, \rho_{\text{assumed}} = 0.1, \rho_{\text{true}} = 0.05$ | MSM (Mismatched) | 38.28 |
>
> > The theoretical analysis assumes i.i.d. sampling of the measurement operators S. In practice, acquisition patterns in MRI, CT, or PET are typically structured and hardware-constrained rather than i.i.d., so the practical validity of this assumption should be discussed.
>
> Our i.i.d. assumption applies only to the algorithmic subsampling masks used within MSM inference—not to the physical acquisition, which may be fully deterministic or hardware-constrained. Thus, our theory does not require MRI/CT/PET hardware to acquire measurements i.i.d. or at random. We clarify this in **Appendix C.1**.

---

> ### Author Response · Authors · 2025-11-20
> **Response to reviewer 8nET (2/2)**
>
> > In Eq. (4), the maximum operation in computing the weighting function is reasoned by avoiding division by 0. However, in my understanding this would anyways be detrimental to the sample quality, if there are regions not covered at all by the generation process. Are there any insights on how many sampling steps and stochastic loop iterations are needed to ensure this?
>
> The scenario of failing to update certain regions during the whole sampling occurs extremely rarely, as the combination of MSM’s stochastic loops and the number of diffusion steps contributes to all measurement coordinates being repeatedly updated.
> Concretely, if $p$ denotes the fraction of coordinates dropped at each stochastic loop, then the probability that a specific coordinate is never selected across all diffusion steps is $p^{\text{num steps} * w}$. This quantity decays exponentially and becomes vanishingly small under typical sampling settings. For example, with $p = 0.4$, 100 diffusion steps, and $w=1$, this probability is $0.4^{100} \approx 1.6 \times 10^{-40}$, effectively zero. This indicates that MSM practically achieves full coverage of the measurement domain during sampling. We have added this clarification in **Appendix C.1**.
>
>
> > Evaluating the FID based on 3k samples (L1126) is a rather low sample number, at least 10k should be used. This is particularly important for unconditional generation, where diversity matters and FID variance can be high. In any case, if the choice is substantially lower than common practice, it should be discussed and justified.
>
> Prompted by the reviewer, we recomputed all unconditional FID scores using 10k generated samples and updated the corresponding tables in **Section 4**, along with the FID computation details in **Appendix D.7**.
> The table below shows that our method remains superior to the baselines under the same training-data settings.
>
> The table shows the newly computed FID on the FFHQ human face dataset.
>
> | *Training data* | *Method* | *FID ↓* |
> |-------------------|------------|-----------|
> | No degradation | Oracle diffusion | 10.21 |
> | $p = 0.4, \rho = 0$ | MSM | **29.14** |
> | $p = 0.4, \rho = 0$ | Ambient diffusion | 55.90 |
> | $p = 0.4, \rho = 0.1$ | MSM | **37.14** |
> | $p = 0.4, \rho = 0.1$ | GSURE diffusion | 89.71 |
>
>
> The table shows the newly computed FID on the fastMRI brain MRI dataset.
>
> | *Training data* | *Method* | *FID ↓* |
> |-------------------|------------|-----------|
> | No degradation | Oracle diffusion | 28.41 |
> | $R = 4, \rho = 0$ | MSM | **64.37** |
> | $R = 4, \rho = 0$ | Ambient diffusion | 70.07 |
> | $R = 4, \rho = 0.1$ | MSM | 82.17 |
>
>
>
> > Given that MSM requires stochastic loops $w$ and aggregation of partial measurement scores, which is not the case for Ambient DPS, a comparison on inference time would be interesting.
>
> Prompted by the reviewer, we have evaluated MSM under multiple choices of the stochastic loop count $w\in {1, 2, 3}$. The table below shows that, even with the minimal setting $w = 1$, MSM continues to outperform Ambient DPS. We have added these results and the corresponding discussion in **Appendix E.10**.
>
> | *Setup* | *Metric* | *Input* | *A-DPS* | *MSM (w = 1)* | *MSM (w = 2)* | *MSM (w = 3)* |
> |-----------|------------|-----------|-----------|------------------|------------------|------------------|
> | *Inpainting* | PSNR ↑ | 18.26 | 20.14 | 24.48 | 24.64 | 24.70 |
> | | SSIM ↑ | 0.749 | 0.621 | 0.868 | 0.867 | 0.868 |
> | | LPIPS ↓ | 0.304 | 0.305 | 0.075 | 0.076 | 0.075 |
> | *SR (×4)* | PSNR ↑ | 23.21 | 22.61 | 28.27 | 28.15 | 28.09 |
> | | SSIM ↑ | 0.728 | 0.702 | 0.872 | 0.870 | 0.869 |
> | | LPIPS ↓ | 0.459 | 0.277 | 0.120 | 0.117 | 0.116 |
> | *CS-MRI (×4)* | PSNR ↑ | 22.75 | 27.28 | 29.12 | 30.50 | 30.71 |
> | | SSIM ↑ | 0.648 | 0.804 | 0.806 | 0.828 | 0.839 |
> | | LPIPS ↓ | 0.306 | 0.173 | 0.172 | 0.154 | 0.145 |
> | *CS-MRI (×6)* | PSNR ↑ | 21.94 | 26.29 | 27.20 | 28.21 | 28.86 |
> | | SSIM ↑ | 0.617 | 0.763 | 0.759 | 0.789 | 0.805 |
> | | LPIPS ↓ | 0.342 | 0.201 | 0.202 | 0.178 | 0.168 |
>
>
> > Minor comments L262: box inpainting is not a degradation operator L166: s is not defined L868: S is not a forward operator, it should say subsampling operator.
>
> Thanks for the helpful comments. We have revised **Sections 3.1, 3.3, and Appendix C.1**.
>
>
>
> [1] Charles M. Stein. Estimation of the mean of a multivariate normal distribution. The Annals of Statistics, pp. 1135–1151, 1981.
>
> [2] Kawar et al, Gsure-based diffusion model training with corrupted data. Transactions on Machine Learning Research, 2024.
>
> [3] Daras et al, Consistent diffusion meets tweedie: Training exact ambient diffusion models with noisy data. ICML, 2024.
>
> [4] Aali et al, Solving inverse problems with score-based generative priors learned from noisy data. In 2023 57th Asilomar Conference on Signals, Systems, and Computers, pp. 837–843. IEEE, 2023.

---

> ### Author Response · Authors · 2025-11-26
> **Gentle Reminder to Reviewer 8nET**
>
> Dear Reviewer 8nET
>
> Thank you again for reviewing our paper. We did our best to address your comments and would appreciate any post-rebuttal feedback. Let us know if any additional details would be helpful in supporting a more positive assessment of our work.
>
> Authors

---

> > ### Comment · Reviewer_8nET · 2025-11-26
> > **Response**
> >
> > Thank you for your detailed response. My concerns are addressed, I will therefore raise my rating to 6.

---

> > > ### Author Response · Authors · 2025-11-26
> > >
> > > We thank the reviewer for all the feedback, engaging in this review, and raising the score. Please let us know if any additional details would be helpful in supporting a more positive assessment of our work.

---

### Author Response · Authors · 2025-11-20
**General response to AC and reviewers**

Dear Reviewers and ACs,

Thank you all for reviewing our work and providing feedback. In response, we revised the manuscript to clarify several conceptual points and added new experimental results. To accommodate these additions, we moved Section 4 (Theoretical Analysis) to Appendix A, while retaining the main theoretical result in Section 3 (Method: Measurement Score-Based Diffusion Model). All newly added experiments and clarifications are incorporated into the paper (highlighted in brown text), and each change is referenced with its exact location in our responses.

Authors

---

### Author Response · Authors · 2025-12-02
**Summary of rebuttal**

Dear Area Chair,

We appreciate reviewing our work and providing feedback. We would like to provide a brief, factual summary of the discussion and revisions for our paper.

All reviewers acknowledged the *novelty and practical relevance* of the MSM framework for learning diffusion-based priors directly in the measurement domain without clean data. Reviewer `hKzq` emphasized the **originality of learning partial measurement scores** and strong experimental validation over self-supervised baselines in *both* unconditional generation and inverse problems. Reviewer `5Qe2` highlighted the well-designed stochastic sampling and aggregation pipeline. Reviewers `8nET` and `a8c4` noted that Theorem 1 provides formal justification for the stochastic approximation, with experimental results confirming its effectiveness in practice.

---

## Summary of Revisions During Discussion

During the rebuttal phase, we addressed **all major technical concerns** through precise theoretical clarifications, new ablation studies, and expanded experimental evaluations. The key issues and corresponding actions are summarized below:
 | **Raised Concern** | **How It Was Addressed** |
|---|---|
| Need for clarification of the relationship between clean-data score and measurement score | Clarified that MSM is **fundamentally distinct from prior baselines**: while baselines approximate the clean-data score, MSM directly models partial-measurement scores as a principled surrogate prior, the **core contribution** supported by theory and SOTA experiments (`Sections 1 and 3.2`). |
| Need to justify why measurement domain is better than corrupted image domain | Clarified that training in the measurement domain removes ambiguity: each subsampled measurement is uniquely defined by the acquisition operator, whereas infinitely many corrupted images can map to the same measurement. It ensures physically well-defined denoising targets (`Section 1`). |
| Scope of applicability beyond RGB inpainting and multi-coil MRI | Clarified broader applicability to sparse-view CT, limited-angle CT, and PET (`Appendix C.1`). |
| Need to demonstrate robustness under training-time noise mismatch | Added mismatch training experiment showing **only minimal degradation** in FID (`Appendix E.9`). |
| Need to demonstrate robustness under inference-time noise in inverse problems | Added inverse-problem evaluations under three different noise levels (0.005, 0.01, 0.02) showing **consistent superiority** over diffusion baselines (`Appendix E.7`). |
| Need for a larger sample size for FID computation (beyond 3k samples) | Recomputed all FID scores with 10k samples; **MSM remains superior** (`Section 4, Appendix D.7`). |
| Need for comparison with baselines under the same computational complexity | Added comparison under matched computational budget showing that **MSM outperforms diffusion baselines** even when using equal or lower NFE (`Appendix E.10`). |


---

## Summary of the Evaluations

| **Reviewer** | **Progress of Discussion** | **Rating Before Discussion** | **Outcome Prior to Reversion** |
|---|---|---|---|
| 8nET | All concerns explicitly resolved | 4 | **Score raised to 6** |
| a8c4 | All technical points formally clarified and revised in the manuscript | 4 | Full response provided; discussion was ongoing when the forum closed |
| hKzq | All requested experiments conducted and clarifications incorporated into the manuscript | 6 | No post-rebuttal response before forum closure |
| 5Qe2 | All requested experiments conducted and clarifications incorporated into the manuscript | 6 | No post-rebuttal response before forum closure |


We respectfully ask that this full discussion context, including the resolved feedback and the raised post-rebuttal score, be taken into careful consideration.

Sincerely,

 Authors

---

### Meta-Review · Area_Chair_ZBcR · 2026-01-09

**Summary:**

The paper studies an interesting problem and proposes the Measurement Score-based Model (MSM) to learn partial measurement scores directly without requiring clean ground-truth images. By aggregating the partial measurement scores in expectation, it allows unconditional generation of full clean data and conditional generation for solving inverse problems. Experiments on unconditional and conditional tasks including inpainting and MRI reconstruction show good performance compared to prior approaches such as ambient diffusion.

Reviewers raise the questions and concerns on the partial measurements score, applicability to other tasks like CT reconstruction, training and sampling robustness, baseline comparison under computational constraints, etc. During the rebuttal, the author provides an extensive response with detailed clarifications and new experiments results to answer these questions.

After the rebuttal, two reviewers responded with follow-up discussion. One reviewer raised the score from 4 to 6 and confirmed the concerns are well addressed, while the other reviewer kept the score and raised follow-up questions on the author’s claim in rebuttal and paper. The subsequent clarifications from the author may be helpful to further address these questions.

Overall, the paper studies an interesting and important problem in diffusion inverse solvers when complete clean data is not available to train the model, and proposes a reasonable methodology. Although there are obvious limitations in the proposed approach, for example, assuming various partial masks to cover the full measurement space which may not be the case in real-world applications, and limited to broader inverse problems such as motion deblurring etc. (the argument on the partial deblurring may not be fully convincing considering more common practical scenarios), the paper may bring an interesting insight and discussion to the community to encourage more research efforts in this direction. Thus, I think the paper is around borderline while leaning towards an acceptance due to the reason above.

**Reviewer Concerns:**

Reviewers raise the questions and concerns on the partial measurements score, applicability to other tasks like CT reconstruction, training and sampling robustness, baseline comparison under computational constraints, etc. During the rebuttal, the author provides an extensive response with detailed clarifications and new experiments results to answer these questions.

**Reviewer Scores:**

After the rebuttal, two reviewers responded with follow-up discussion. One reviewer raised the score from 4 to 6 and confirmed the concerns are well addressed, while the other reviewer kept the score and raised follow-up questions on the author’s claim in rebuttal and paper. The subsequent clarifications from the author may be helpful to further address these questions.

---

### Decision · Program_Chairs · 2026-01-26

Accept (Poster)